# Influence of the dry aerosol particle size distribution and morphology on the cloud condensation nuclei activation. An experimental and theoretical investigation

Junteng Wu[1], Alessandro Faccinetto[1], Symphorien Grimonprez[1], Sébastien Batut[1], Jérôme Yon[2], Pascale Desgroux[1], Denis Petitprez[1]

[1]Lille Univ., CNRS PC2A, 59000 Lille, France
[2]Normandie Univ., INSA Rouen, UNIROUEN, CNRS CORIA, 76000 Rouen, France

*Correspondence to*: Alessandro Faccinetto (alessandro.faccinetto@univ-lille.fr)

**Abstract.** Combustion and other high temperature processes frequently result in the emission of aerosols in the form of polydisperse fractal-like aggregates made of condensed phase nanoparticles (soot for instance). If certain conditions are met, the emitted aerosol particles are known to evolve into important cloud condensation nuclei (CCN) in the atmosphere. In this work, the hygroscopic parameter $\kappa$ of complex morphology aggregates is calculated from the supersaturation dependent activated fraction $F_a = F_a(SS)$ in the frame of $\kappa$-Köhler theory. The particle size distribution is approximated with the morphology-corrected volume equivalent diameter calculated from the electrical mobility diameter by taking into account the diameter of the primary particle and the fractal dimension of the aggregate experimentally obtained from transmission electron microscopy measurements. Activation experiments are performed in water supersaturation conditions using a commercial CCN-100 condensation nuclei counter. The model is tested in close-to-ideal conditions of size selected, isolated spherical particles (ammonium sulfate nanoparticles dispersed in nitrogen), then with complex polydisperse fractal-like aggregates (soot particles activated by exposure to ozone with $\kappa$ as low as $5\times10^{-5}$) that represent realistic anthropogenic emissions in the atmosphere.

**Keywords**. $\kappa$-Köhler theory, cloud condensation nuclei (CCN), size distribution, morphology, hygroscopic parameter, soot.

## 1. Introduction

Soot particles formed during the incomplete combustion of hydrocarbons and emitted in the exhausts are potentially important contributors to the radiative forcing of the atmosphere as they adsorb and scatter the solar radiation (direct effect), but can also act as cloud condensation nuclei (CCN) or ice nuclei (IN) and trigger the formation of persistent clouds (indirect effect) (Bond et al., 2013). To date, estimations over the magnitude of the direct and indirect effects are subject to large uncertainty, and commonly accepted values span the range [+0.25, +1.09] W m$^{-2}$ for the direct effect and [-1.20, 0.00] W m$^{-2}$ for the indirect effect comprehensive of all aerosol-cloud interactions (Stocker et al., 2014). Such large uncertainties result from the combination of several difficult-to-predict behaviors of the soot particles in the atmosphere when compared to mineral and biogenic aerosols. For instance, their small size and low density enable for a long lifetime in the atmosphere that can reach several weeks (Govardhan et al., 2017). Their complex morphology and large specific surface allow many possible surface interactions that can deeply affect their reactivity (Monge et al., 2010; Browne et al., 2015). Furthermore, their number concentration is subject to high geographic variability, and especially in polluted regions can be comparable to the typical number concentration of marine aerosols that provide the largest contribution to the total mass of aerosols emissions (Rose et al., 2006; Sayer et al., 2012). To better understand the effect of soot particle on the radiative balance of the atmosphere, it is therefore important to understand how their size distribution, morphology, and surface composition impact their activity as CCN.

Soot formation in flame combustion is a complex process strongly affected, among other factors, by the fuel nature, the local fuel-air equivalence ratio and the flame temperature (D'Anna, 2009; Wang, 2011; Karataş and Gülder, 2012). At short reaction time, or equivalently at low height above the burner (HAB), the oxidation of the fuel generates small radicals that quickly recombine to form larger and larger hydrocarbons, molecular ions and radicals. Small polycyclic aromatic hydrocarbons (PAHs) and their derivatives are since long regarded as the most important soot molecular precursors (Richter and Howard, 2000). At longer reaction time, or equivalently at higher HAB, the soot molecular precursors react to form condensed phase nascent soot particles. In controlled, low sooting laboratory flames, nascent soot particles as small as 2 nm can be detected (Betrancourt et al., 2017). The dynamic equilibrium between the heterogeneous reactions at the particle surface (surface growth and oxidation) in concert with coalescence and coagulation phenomena determine whether the newly formed particles increase their size or are re-oxidized into gas phase products. In sooting flames, nascent soot particles quickly evolve into primary soot particles having typical diameter in the range 5-30 nm (Apicella et al., 2015). If their number concentration is sufficiently large, condensed phase particles at any reaction time can coalesce and coagulate into complex morphology aggregates that persist up to the flame exhausts (Kholghy et al., 2013).

A substantial body of literature exists on the characterization of the morphology of soot particles by electron microscopy. Young primary soot particles sampled at low HAB are well known to be characterized by an amorphous core surrounded by a highly structured series of concentric shells, often referred to as onion-like structure (Kholghy et al., 2016). On the other hand, mature soot particles sampled at high HAB tend to be extremely complex aggregates of hundreds to tens of thousands of primary particles (Santamaría et al., 2007; Kelesidis et al., 2017). The morphology of these aggregates is scale invariant over a relatively large range (*fractal-like* aggregates), and therefore some concepts borrowed from fractal geometry can be applied to characterize them. In particular, the fractal dimension $D_f$ is considered to be an important descriptor of the soot particles morphology that links the number of primary particles of an aggregate $N_{pp}$ to the diameter of the primary particle $d_{pp}$ through the power law (Sorensen, 2011; Eggersdorfer and Pratsinis, 2014):

$$N_{pp} = k_f \left( \frac{d_p}{d_{pp}} \right)^{D_f}$$

Eq. (1)

where $k_f$ is the exponential pre-factor and $d_p$ the equivalent particle diameter. Over the years, different approaches have been proposed to estimate $d_p$ with quantities easily accessible from experiments that include the diameter of gyration from angular light scattering measurements (Köylü et al., 1995; Sorensen et al., 1992) or the size of the aggregate projection from scanning electron microscopy (Colbeck et al., 1997) and transmission electron microscopy (Cai et al., 1995; Hu et al., 2003).

From the chemical point of view, the gas-condensed phase conversion remains to date a poorly understood process (D'Anna, 2009; Wang, 2011; Desgroux et al., 2013; Michelsen, 2017). The molecular precursors participating to soot formation and found adsorbed on the surface of soot particles can be as small as PAHs containing 3-7 aromatic cycles, or as large as tens of aromatic cycles depending on the combustion conditions (Irimiea et al., 2019). The availability of surface hydrogen atoms is considered to be a driving force of the surface growth process and is often described by the hydrogen abstraction/acetylene addition mechanism (Frenklach and Wang, 1990; Frenklach, 2002). At the particle surface, reactive young soot is generally rich of small PAHs characterized by high H/C ratio (> 0.7), in contrast to more inert mature soot that is characterized by low H/C ratio (< 0.4). Being PAHs thermodynamically stable compounds, they are often found adsorbed on the surface of the soot particles in the exhausts and give a significant contribution to the soot particles reactivity.

Such a large variability of size distribution, morphology and chemical composition strongly impacts the reactivity of soot particles in the atmosphere and their propensity to evolve into CCN. Several studies exist that characterize the CCN activity of soot particles generated in the exhausts of laboratory flames (Lambe et al., 2015) and commercial burners like the miniCAST (Henning et al., 2012; Friebel et al., 2019). Soot particle aging experiments are often performed in laboratory conditions that simulate the atmosphere and make use of flow reactors (Kotzick et al., 1997; Lambe et al., 2015; Zuberi et al., 2005) or atmospheric simulation chambers (Tritscher et al., 2011; Wittbom et al., 2014; Grimonprez et al., 2018). The hygroscopic properties of soot are generally determined at supersaturation conditions provided by instruments such as variable supersaturation condensation nuclei counters (VSCNC) or cloud condensation nuclei counters (CCNc). Overall, freshly emitted soot particles are generally considered as poor CCN. However, several studies demonstrate that photochemical aging (Tritscher et al., 2011) or chemical aging that includes exposition to OH radicals (Zuberi et al., 2005; Lambe et al., 2015), to $O_3$ (Kotzick et al., 1997; Wittbom et al., 2014; Grimonprez et al., 2018) or to $NO_3$ radicals (Zuberi et al., 2005) under atmospheric relevant conditions can efficiently turn soot particles into CCN.

Köhler theory (Köhler, 1936) is widely used to describe the formation process of liquid cloud droplets at supersaturation conditions. Köhler theory is entirely founded on equilibrium thermodynamics, and describes the change of the saturation vapor pressure of water induced by the curved surface of the nascent droplet and by the presence of solutes in the liquid phase. A number of recent implementations of Köhler theory have been used to describe the cloud droplet activation of wettable insoluble or partially soluble compounds. Among them, the adsorption activation theory describes the mechanism of droplet growth through multilayer adsorption of water. The number of layers of adsorbed water molecules is calculated using Brunauer, Emmet and Teller isotherms (Henson, 2007), or alternatively the Frenkel-Halsey-Hill isotherms (Sorjamaa and Laaksonen, 2007). In this work, a simpler approach is chosen (Petters and Kreidenweis, 2007) that relies on a single parameter ($\kappa$) representation of the CCN activity to take into account the reduction of the water activity due to the presence of partially soluble components ($\kappa$-Köhler theory). According to $\kappa$-Köhler theory, at thermodynamic equilibrium the supersaturation over an aqueous solution droplet $SS = SS(D, d_p, \kappa)$ as a function of the droplet diameter $D$, of the size of the seeding particle $d_p$ and of the hygroscopic parameter $\kappa$ is given by:

$$SS(D, d_p, \kappa) = \frac{D^3 - d_p^3}{D^3 - d_p^3(1-\kappa)} \exp\left(\frac{A}{D}\right) - 1, \qquad A = \frac{4 M_w \sigma_{s/a}}{R\, T\, \rho_w} \qquad \text{Eq. (2)}$$

where $M_\text{w}$ and $\rho_\text{w}$ are the molar mass and density of water, respectively, $\sigma_\text{s/a}$ is the surface tension at the solution/air interface, $R$ is the ideal gas constant and $T$ is the temperature. $\sigma_\text{s/a}$ = 0.072 J m$^{-2}$ and $T$ = 298 K are commonly used for calculations that lead to $A \simeq 2.09 \times 10^{-9}$ m.

Even if this approach does not describe explicitly the underlying mechanism of CCN formation, the $\kappa$-Köhler theory has been widely used to characterize the activity in supersaturation conditions of isolated non-spherical aerosol particles as a function of $\kappa$ (Su et al., 2010; Cerully et al., 2011), $d_\text{p}$ (Snider et al., 2006; Kuwata and Kondo, 2008), or $SS$ (Sullivan et al., 2009; Tang et al., 2015). The particle size distribution has been proven to affect the CCN activation curves (Abdul-Razzak and Ghan, 2000; Snider et al., 2006). However, the (geometric) standard deviation of the particle size distribution alone is not sufficient to completely explain the slope of the CCN activation curves (Snider et al., 2006), and therefore a distribution of values of the parameter $\kappa$ has been proposed to add the missing degree of freedom (Su et al., 2010; Cerully et al., 2011).

Only a few studies exist on the CCN activity of soot particles compared to non-aggregated aerosol particles (Grimonprez et al., 2018; Lambe et al., 2015; Sullivan et al., 2009; Tang et al., 2015). In these studies, a corrected volume equivalent diameter based on the estimated mass density of the soot particles is used to parameterize the particle size. However, particles characterized by very low activity like fresh soot particles have been reported to have $\kappa < 0$ ("apparent crossing of the Kelvin limit") in some instances (Grimonprez et al., 2018; Lambe et al., 2015; Tritscher et al., 2011). To avoid this problem, $\kappa$ has been obtained from the fitting of the activation curve with generic sigmoid functions that do not take into account the particle size distribution or a distribution of values of $\kappa$. In the above mentioned studies, both for non-aggregated and aggregated aerosol particles, the electrical mobility diameter, experimentally accessed by scanning mobility particle sizing (SMPS), has often been used to measure the particle size distribution. In this case, the role of multiply charged particles needs to be taken into account (Petters, 2018).

The main goal of this work is the quantification of the role of the particle morphology on the cloud condensation activity of soot particles in conditions that simulate atmospheric chemical aging. In practice, a morphology-corrected volume equivalent diameter $d_\text{ve}$ is calculated from the electrical mobility diameter $d_\text{m}$ by including the diameter of the primary particle $d_\text{pp}$ and the fractal dimension $D_\text{f}$ obtained from transmission electron microscopy (TEM). $\kappa$ is calculated from the best fit of the experimental activation data $F_\text{a} = F_\text{a}(SS)$ obtained in water supersaturation conditions using a commercial CCN-100 condensation nuclei counter. After including the contribution of the morphology in $d_\text{ve}$, $\kappa$ is considered to be only representative of the particle chemistry such as the modification of the surface composition or the formation of soluble compounds due to the chemical aging. To account for the heterogeneity of the particle chemistry and to correct for the differences between experimental and calculated $F_\text{a} = F_\text{a}(SS)$, $\kappa$ is treated as a probability distribution rather than a single value. The second main goal of this work is to provide quantitative information on the evolution of $\kappa$ during the chemical aging of soot particles characterized by different maturity, i.e. sampled at different HAB in a laboratory jet diffusion flame supplied with kerosene. The model is first tested in close-to-ideal conditions of size selected, isolated spherical particles (ammonium sulfate nanoparticles dispersed in nitrogen), then in the complex case of polydisperse fractal-like aggregates (soot particles activated by exposure to ozone).

## 2. Theory

### 2.1. Modification of $F_a(SS)$ to include a distributions of $d_p$ and $\kappa$

Probability density functions of $d_p$ (Abdul-Razzak and Ghan, 2000; Snider et al., 2006) and $\kappa$ (Cerully et al., 2011; Su et al., 2010) have been widely used in aerosol science and atmospheric research to describe the CCN activity. Lognormal distributions are considered as viable approximations for $p(\kappa)$, while $d_p$ and $\kappa$ are often treated as uncorrelated variables to avoid double integration (Su et al., 2010; Zhao et al., 2015). The activated fraction $F_a(SS)$ is then calculated as:

$$F_a(SS) = \sum_{\kappa=0}^{\infty} \left\{ \frac{1}{2} - \frac{1}{2} \text{erf} \left[ \frac{\ln d_p(\kappa, SS) - \ln \mu_{p,geo}}{\sqrt{2} \ln \sigma_{p,geo}} \right] \right\} p(\kappa) \Delta\kappa \qquad \text{Eq. (3)}$$

where $\mu_{p,geo}$ and $\sigma_{p,geo}$ are the geometric mean (i.e. the median) and the geometric standard deviation of $d_p$, respectively. $p(\kappa)$ is the probability density function of $\kappa$:

$$p(\kappa) = \frac{1}{\kappa \ln \sigma_{\kappa,geo} \sqrt{2\pi}} e^{-\frac{[\ln \kappa - \ln \mu_{\kappa,geo}]^2}{2\ln^2 \sigma_{\kappa,geo}}} \qquad \text{Eq. (4)}$$

where $\mu_{\kappa,geo}$ and $\sigma_{\kappa,geo}$ are the geometric mean (i.e. the median) and the geometric standard deviation of $\kappa$, respectively.

### 2.2. Definition of the morphology-corrected volume equivalent diameter $d_{ve}$

In this section, a relationship to obtain $d_{ve}$ from $d_m$ for a fractal-like aggregate is derived. $d_{ve}$ is the diameter of a sphere having the same volume as the aggregate, and assuming the aggregate made of identical, spherical primary particles, is defined as:

$$d_{ve} = d_{pp} N_{pp}^{\frac{1}{3}} \qquad \text{Eq. (5)}$$

where $d_{pp}$ and $N_{pp}$ are the diameter and number of primary particles per aggregate, respectively. It is worth to notice that often, for practical purposes, the value of $d_{pp}$ used in calculations is the mass equivalent diameter of the primary particle distribution obtained from TEM measurements. On the other hand, $d_m$ is directly linked to the aerodynamic force acting on the particle $F_{drag}$ (Dahneke, 1973; Tritscher et al., 2011) and can be directly obtained from SMPS measurements:

$$F_{drag} = \frac{3\pi \eta d_m v_r}{C_c(d_m)} \qquad \text{Eq. (6)}$$

where $\eta$ and $v_r$ are the kinematic viscosity of the gas and the particle-gas relative velocity, and $C_c$ is the Cunningham slip factor (Allen and Raabe, 1985):

$$C_c(K_n) = 1 + K_n \left[ 1.142 + 0.558 \exp\left(-\frac{0.999}{K_n}\right) \right] \qquad \text{Eq. (7)}$$

$K_n = 2\lambda_g/d_m$ is the Knudsen number and $\lambda_g$ is the gas mean free path. The drag force acting on an aggregate $F_{drag,agg}$ can be approximated using the drag force acting on each primary particle $F_{drag,pp}$, which is considered as a sphere, using the relation (Yon et al., 2015):

$$F_{drag,agg} = F_{drag,pp} N_{pp}^{\frac{\Gamma}{D_f}} \qquad \text{Eq. (8)}$$

The exponential factor $\Gamma = \Gamma(d_{pp})$ has been empirically estimated as a function of the Knudsen number (Yon et al., 2015) for soot particles generated with a miniCAST commercial burner (propane-air diffusion flame). In the range 1.61 < $D_f$ <1.79:

$$\Gamma = 1.378 \left[\frac{1}{2} + \frac{1}{2}\text{erf}\left(\frac{K_n(d_{pp}) + 4.454}{10.628}\right)\right] \qquad \text{Eq. (9)}$$

Although the variability range of $D_f$ might seem quite restrictive, in practice it covers a region representative of soot aggregates (Yon et al., 2015; Kelesidis et al., 2017). Therefore, we make the additional hypothesis that Eq. (9) can be applied to a variety of experimental investigations including our case. Introducing Eq. (6) in Eq. (8) yields:

$$d_m = \frac{C_c(d_m)}{C_c(d_{pp})} \ d_{pp} N_{pp}^{\frac{\Gamma}{D_f}} \qquad \text{Eq. (10)}$$

The dependence on $N_{pp}$ can be removed by using the definition of $d_{ve}$ in Eq. (5). Finally, Eq. (10) can be solved for $d_{ve}$ to yield:

$$d_{ve}(d_{pp}, D_f, d_m) = d_{pp}\left[\frac{d_m}{d_{pp}}\frac{C_c(d_{pp})}{C_c(d_m)}\right]^{\frac{D_f}{3\Gamma}} \qquad \text{Eq. (11)}$$

$d_{ve} = d_{ve}(d_{pp}, D_f, d_m)$ can be calculated once size distribution and morphology of the aerosol are known using Eq. (11). $d_{pp}$ and $D_f$ are obtained from TEM imaging as explained below, while $d_m$ is easy to access from SMPS measurements.

Fig. 1 shows the functional dependency of $d_{ve}$ on (a) $d_{pp}$ and (b) $D_f$. As an example, the experimental $d_m$ distribution of soot particles after size selection at 150 nm (black solid line, see section 3 for details on the experimental conditions) is compared to calculated $d_{ve}$ (colored dashed and dotted lines). $d_m$ is measured immediately before the droplet nucleation experiments in tandem SMPS configuration. In the cases investigated in this work, $d_{ve} = d_{ve}(d_{pp}, D_f, d_m)$ is always significantly shifted to smaller values and narrower than the original $d_m$. Increasing $d_{pp}$ from 10 nm up to 30 nm ($D_f$ = 1.7) results in $d_{ve}$ increasing from 82.0 nm up to 117.0 nm. Similarly, increasing $D_f$ from 1.6 up to 1.8 ($d_{pp}$ = 20 nm) results in $d_{ve}$ increasing from 93.5 nm up to 113.3 nm.

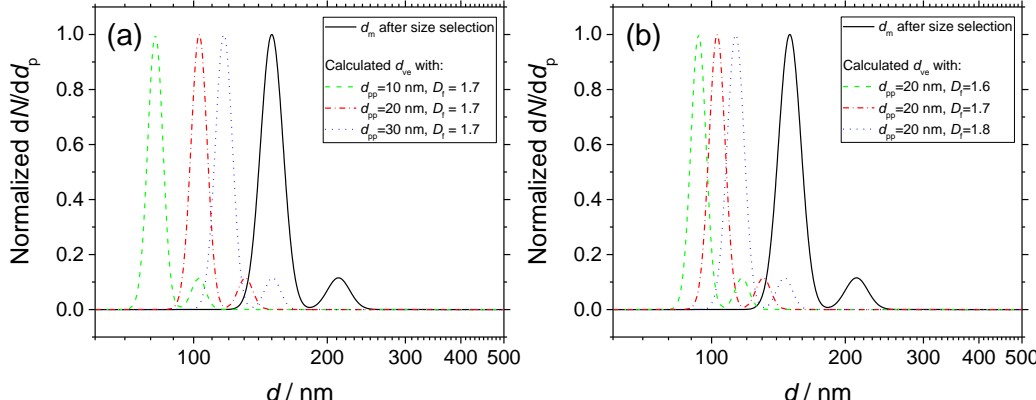

**Fig. 1.** $d_\mathrm{m}$ distribution of soot particles sampled from the kerosene jet diffusion flame at 130 mm HAB and size selected at 150 nm (black solid line) measured immediately before droplet nucleation experiments. Simulations of $d_\mathrm{ve} = d_\mathrm{ve}(d_\mathrm{pp}, D_\mathrm{f}, d_\mathrm{m})$ of soot particles having complex morphology according to Eq. (11).. For each series of simulations (colored dashed and dotted lines), (a) $d_\mathrm{m}$ and $D_\mathrm{f}$, or alternatively (b) $d_\mathrm{m}$ and $d_\mathrm{pp}$ are set as constant and the remaining parameter is varied in the range indicated in the legend.

## 2.3. Taking into account the multimodality of $d_\mathrm{m}$

Because of the existence of multiple charges, one lognormal fit is generally not sufficient to describe the $d_\mathrm{m}$ distributions. Therefore, with the aim of maintaining an analytical approach, an empirical multi-mode lognormal fit is used as a simple solution to describe the $d_\mathrm{m}$ distribution of the aerosol particles injected in the CCNc:

$$\frac{\mathrm{d}N(d_\mathrm{p})}{\mathrm{d}d_\mathrm{p}} = \sum_i \frac{N_i}{d_\mathrm{p} \ln \sigma_{\mathrm{p,geo,i}} \sqrt{2\pi}} e^{-\frac{[\ln d_\mathrm{p} - \ln \mu_{\mathrm{p,geo,i}}]^2}{2 \ln^2 \sigma_{\mathrm{p,geo,i}}}} \qquad \text{Eq. (12)}$$

where $\mu_{\mathrm{p,geo,i}}$ and $\sigma_{\mathrm{p,geo,i}}$ are the geometric mean and standard deviation of each mode.

## 3. Experimental approach

In this section, the methodological approach is described. An overview of the experimental aerosol generation setup is shown in Fig. 2(a) for ammonium sulfate, and (b) for soot particles. Ammonium sulfate represents the simplest case that is well known in the literature (Rose et al., 2008) for the isolated, quasi-spherical particles that can be generated by atomization (section 3.1). Freshly generated ammonium sulfate aerosols are size selected then injected in a 50 L Pyrex glass static reactor. From the reactor, particles are sampled for activation experiments, size and morphology measurements. Soot particles are sampled from a jet diffusion flame supplied with kerosene, then chemically aged with ozone (section 3.2). All activation experiments are performed in a CCN-100 commercial nucleation chamber (section 3.3). The particle size and morphology are characterized by SMPS and TEM, respectively (section 3.4).

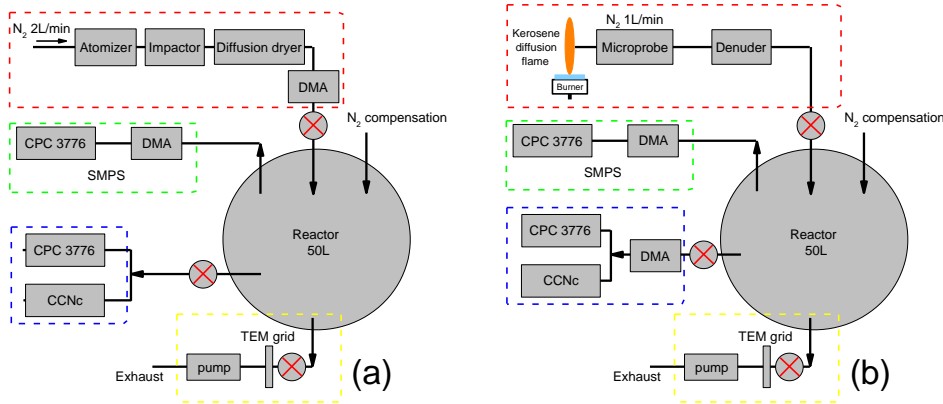

**Fig. 2. Overview of the experimental setup used for (a) ammonium sulfate and (b) soot. Aerosol injection system (red frame), size distribution measurement system (green frame), activated fraction measurements system (blue frame), and collection system for TEM grids (yellow frame).**

### 3.1. Ammonium sulfate aerosols generation

Ammonium sulfate aerosols are generated by atomization using a TSI aerosol generator 3076 loaded with 0.1 g L$^{-1}$ concentration aqueous solution (98% purity ammonium sulfate purchased from Sigma Aldrich and 18.2 MΩ cm purity water produced in a laboratory VEOLIA BIOPURE 15 apparatus). Nitrogen is used for
atomization, in the range 2-3 L min$^{-1}$ depending on the target aerosol concentration. The excess water is removed by flowing the aerosol in a 60 cm long diffusion dryer loaded with silica gel orange.

### 3.2. Soot aerosol generation, sampling and aging

A detailed description of the burner and sampling system is given elsewhere (Grimonprez et al., 2018). Briefly, a turbulent kerosene jet flame is stabilized on a Holthuis burner modified to allow the installation of a direct
injection high efficiency nebulizer. The high speed spray of liquid fuel droplets at the exit of the nebulizer is ignited by a pilot methane flat flame on the outer ring of the modified burner, resulting in a turbulent diffusion flame approximately 21 cm high. All experiments are performed using kerosene fuel Jet A-1. Soot is extracted from the jet flame at 70 and 130 mm HAB. At 70 mm HAB the particle concentration is small and the gas phase is rich in condensable hydrocarbons. At higher HAB, the particle concentration increases and both the diameter
of the primary particles and the mobility diameter of the aggregates grow due to surface reaction and coagulation. This results in the increase of the soot volume fraction up to a peak at 130 mm HAB. Above 210 mm HAB all the particles are oxidized resulting in a non-smoking flame. Soot is sampled with a diluting quartz microprobe. The sampled flow is analyzed online by SMPS or deposited on Lacey grid for TEM analyses. This setup allows a fast dilution of the sampled gas up to a factor 3×10$^4$ that quenches most chemical reactions
and limits particle coagulation and aggregation downstream in the sampling line. It is important to notice that the particle concentration in the sampling line has to be larger than the optical counters detection limit but low enough to limit post-sampling aggregation or agglomeration. To avoid the formation of secondary organic aerosol in the reactor, a parallel plate, activated carbon diffusion denuder is installed downstream the microprobe.

Freshly emitted soot particles are highly hydrophobic. In this work, the reaction of freshly sampled soot with ozone is used to increase their hydroscopicity following the experimental procedure detailed in our past work

(Grimonprez et al., 2018). The experimental variable used to control the surface oxidation of soot particle is the ozone exposure, defined as the product of the ozone concentration and the residence time in the reactor. Briefly, the reactor is first pumped to reduce the background particle count below the lower detection limit of the CPC, and then is filled with nitrogen and ozone generated by photolysis of oxygen with a UVP SOG-2 lamp. To inject the soot aerosol, the pressure in the reactor is set to a slightly lower value than the sampling line ($\Delta p \simeq -20$ mbar). Therefore, a net sample flow (estimated around 2-5 mL min$^{-1}$) enters the probe from the flame, is immediately mixed with the nitrogen dilution flow (1-8 L min$^{-1}$), passes through the denuder and finally arrives to the reactor. The time origin for the calculation of ozone exposure starts 10 s after the end of soot injection. During injections, the SMPS is disconnected from the reactor to avoid acquiring data outside the recommended pressure range, and reconnected after the pressure has been raised again up to $p$ = 1 bar.

### 3.3. Activation experiments

Activation experiments aim to measure $F_a = F_a(SS)$ and are performed by means of a commercial Droplet Measurement Technologies Cloud Condensation Nuclei counter CCNc-100 installed in parallel to a TSI condensation particle counter CPC 3776. To study the effect of the particle size distribution on $F_a$, two different protocols are adopted for ammonium sulfate and soot. Ammonium sulfate particles are size selected by a DMA, pass through the reactor and then are injected in the CCNc-100. Soot particles are first sampled from the flame and injected in the reactor in which they are aged with ozone (section 3.2). A TSI differential mobility analyzer DMA 3081 is installed immediately downstream the reactor so that only aerosol particles with a selected electrical mobility and geometric deviation are injected in the CCNc-100. Additional verifications of the size distribution are performed by SMPS at regular time intervals to rule out the presence of coagulation during the aging experiments. More in detail, a 0.8 L min$^{-1}$ particle-laden flow is sampled from the reactor, split and used to supply the CCNc (0.5 L min$^{-1}$) and the CPC (0.3 L min$^{-1}$) in parallel that record the concentration of nucleated water droplets and aerosol particles at different supersaturations, respectively, required to plot $F_a = F_a(SS)$. The total flow sampled by the CPC and the CCNc is balanced with nitrogen injected directly into the reactor, and all concentrations are corrected for dilution during measurements. Samples on TEM grids are also collected to get information on the morphology and primary particle size distribution of the test aerosol. All activation experiments are taken from our previous work (Grimonprez et al., 2018).

### 3.4. Diagnostics

SMPS measurements are performed to measure the aerosol electrical mobility diameter $d_m$ using a TSI 3091 SMPS that consists of a TSI 3080 DMA upstream a TSI 3776 CPC operated with 0.3 L min$^{-1}$ aerosol flow rate and 1:10 sample/sheath flow ratio. Charge neutralization is provided by a TSI 3088 soft X-ray Neutralizer.

TEM is used to measure $d_{pp}$ and $N_{pp}$ from which $D_f$ is calculated (section 4.2). All parameters are estimated from the TEM images by using ImageJ freeware software. TEM measurements are performed on the FEI Tecnai G2 20 microscope (200 kV acceleration voltage) available at the center for electron microscopy of Lille University. All samples are deposited on Lacey carbon meshes.

## 4. Results and discussion

### 4.1. Test with ammonium sulfate

Ammonium sulfate is well known for the isolated, quasi-spherical particles that can be generated by atomization of aqueous solution, and for this very reason is often used as a reference material for activation experiments (Petters and Kreidenweis, 2007; Rose et al., 2008), and in this work to test the validity of Eq. (3) before moving to complex morphology aggregates. An example of TEM image of the ammonium sulfate particles obtained after size selection is shown in Fig. 3(a), and the corresponding particle projection $d_p$ (TEM) and electrical mobility $d_m$ (SMPS) distributions are shown in Fig. 3(b) and (c), respectively. A single-mode lognormal function is sufficient to fit $d_p$ (black dashed line). However, because of the unavoidable generation of multiple charges during aerosol neutralization, Eq. (12) has to be used to accurately fit $d_m$ (red solid line). As shown in Fig. 3(c), the best result is obtained with a two-mode fit (dotted blue and dashed green lines). The $d_p$ and $d_m$ distributions are representative of the size-selected distribution of ammonium sulfate particles injected in the CCNc.

For isolated, spherical and homogeneous particles, $d_{ve} = d_m = d_p$ (Eggersdorfer and Pratsinis, 2014; Sorensen, 2011). Although Fig. 3 shows a particularly favorable case, in this work it is found that $d_m = d_p$ is always true within 20% uncertainty. A summary of the parameters of the distributions is given in Table 1.

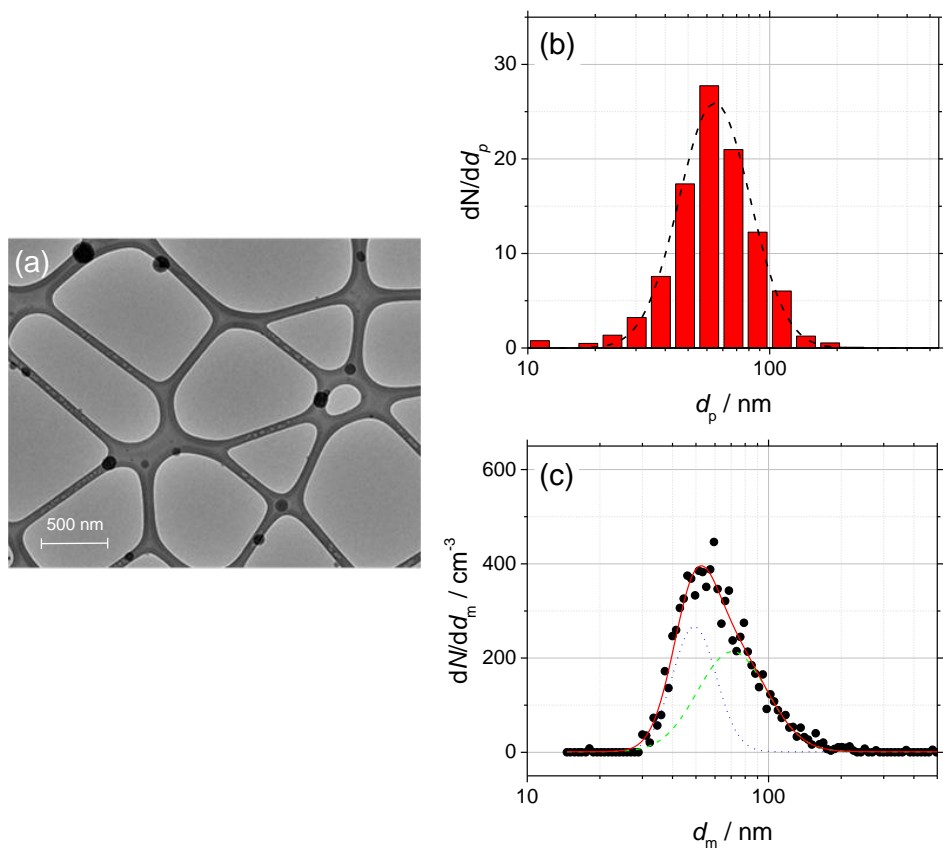

**Fig. 3. (a)** TEM image of size selected ammonium sulfate particles (black quasi-spherical particles) deposited on a Lacey mesh, 6500 magnification. **(b)** diameter of the particle projection $d_p$ obtained from TEM measurements (red bars) and single-mode lognormal fit (black dashed line). **(c)** electrical mobility diameter $d_m$ obtained from SMPS measurements (black dots), and two-mode lognormal fit (red solid line) showing the two contributions (blue dotted and green dashed lines) according to Eq. (12)..

The experimental activation data (black data points) and the calculated activation curves are shown in Fig. 4: $\mu_{p,geo}$ and $\sigma_{p,geo}$ are used as input parameters and obtained independently from SMPS ($d_m$, red solid line) and TEM ($d_p$, black dashed lines) measurements as shown in Fig. 3. Globally, the experimental data are in good agreement with the calculated curves. In the fit, $\mu_{\kappa,geo}$ is set as free parameter while $\sigma_{\kappa,geo}$ is forced to unit value as no variability in the chemical composition of ammonium sulfate is expected. A detailed discussion on the impact of $\sigma_{\kappa,geo}$ on the calculation of $F_a = F_a(SS)$ can be found in Annex A. The calculations from the independently obtained $d_m$ and $d_p$ result in close $p(\kappa)$ having geometric mean $\kappa_{SMPS}$ = 0.58±0.02 and $\kappa_{TEM}$ = 0.61±0.02, both in excellent agreement with $\kappa$ = 0.61 found in the literature (Petters and Kreidenweis, 2007).

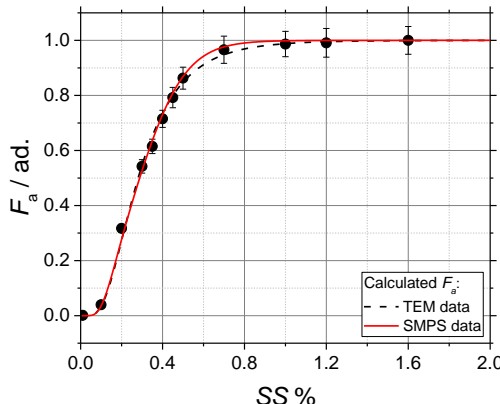

**Fig. 4. Dry ammonium sulfate particles:** activation data obtained from CCNc experiments (black dots) and calculated $F_a = F_a(SS)$ using SMPS (red solid line) and TEM (black dashed lines) data as $d_{ve}$.

| $d_m$ / nm | $d_p$ / nm | $d_m/d_p$ | $\kappa_{SMPS}$ / ad. | $\kappa_{TEM}$ / ad. |
|---|---|---|---|---|
| (mode, SMPS) | (mode, TEM) | (expected: 1.0) | (geometric mean) | (geometric mean) |
| 57.3±0.6 | 59.2±0.8 | 0.97 | 0.58±0.02 | 0.61±0.02 |

**Table 1. Parameters of the lognormal distributions used in the activation experiments of ammonium sulfate particles and shown in Fig. 3. The uncertainties are calculated from the lognormal fit of the distributions. The table also shows the comparison of $\kappa_{SMPS}$ and $\kappa_{TEM}$.**

### 4.2. Characterization of the soot particles morphology

As mentioned above, $d_{pp}$ and $D_f$ of soot aggregates are obtained at the same time from the analysis of TEM images. It is important to notice that Eq. (8) is derived for the mass equivalent diameter and not for the diameter of the projected image (Yon et al., 2015). However, if the soot particles can be considered homogeneous, the mass equivalent diameter and the diameter of the projected image only differ by a scale factor that is easily accounted for.

In this work, $d_{pp}$ is directly obtained by manual counting of the projected images of the soot primary particles.

$D_f$ has been obtained in the past by using at least three different experimental techniques: angular light scattering (Sorensen et al., 1992), scanning electron microscopy (Colbeck et al., 1997) and transmission electron microscopy (Köylü and Faeth, 1992; Cai et al., 1995; Hu et al., 2003). In this work, $D_f$ is estimated from the images of soot aggregates by measuring the maximum length of the aggregate projection $L_{2D}$ (Cai et al., 1995; Köylü et al., 1995):

$$\ln(N_{pp}) = \ln(k_g) + D_f \ln\left(\frac{L_{2D}}{d_{pp}}\right)$$
Eq. (13)

where $k_g$ is the coefficient of the maximum length of the aggregate projection that is treated as a free parameter. As a faster alternative to manual counting, it is possible to estimate $N_{pp}$ from the same set of data used to measure $D_f$ with Eq. (13) by also measuring the total surface of the aggregate projection $A_{2D}$ (Köylü et al., 1995):

$$\ln(N_{pp}) = \ln(k_a) + \alpha \ln\left(\frac{4A_{2D}}{\pi d_{pp}^2}\right)$$
Eq. (14)

where $k_a$ and $\alpha$ are the coefficient and exponent of the projection area, respectively, that for soot fractal aggregates can be either calculated (Medalia, 1967) or measured (Samson et al., 1987; Köylü et al., 1995). $D_f$ of fresh soot particles measured by following this approach (1.665 at 70 mm HAB and 1.647 at 130 mm HAB) is consistent with typical values found in the literature (Samson et al., 1987; Cai et al., 1995; Köylü et al., 1995; Sorensen and Roberts, 1997; Tian et al., 2006). The analysis of TEM images of soot aggregates before and after exposure to ozone confirms that the fractal dimension does not change significantly during the chemical aging process.

The complete characterization of the size and morphology of young soot particles sampled at 70 mm HAB and mature soot particles sampled at 130 mm are shown in Fig. 5 and Fig. 6, respectively. (a) TEM pictures of one aggregate; (b) $d_{pp}$ size distributions; (c) $\ln(N_{pp})$ vs. $\ln(L_{2D}/d_{pp})$ plots from which $D_f$ is obtained according to Eq. (13); (d) SMPS data (black points) and mobility distribution of the soot particles after size selection (black dashed line), and morphology-corrected $d_{ve}$ (red solid line) calculated using Eq. (11). $d_m$ measured in SMPS tandem configuration, as well as $d_{ve}$, are characterized by two well separated modes that can be accurately fitted using Eq. (12). Although not true in general, in the specific case of soot the two modes can be attributed to particles charged +1 (main mode), and particles charged +2 after passing through the first neutralizer that are re-charged +1 after passing through the second neutralizer (smaller mode shifted to the right in the $d$ axis).

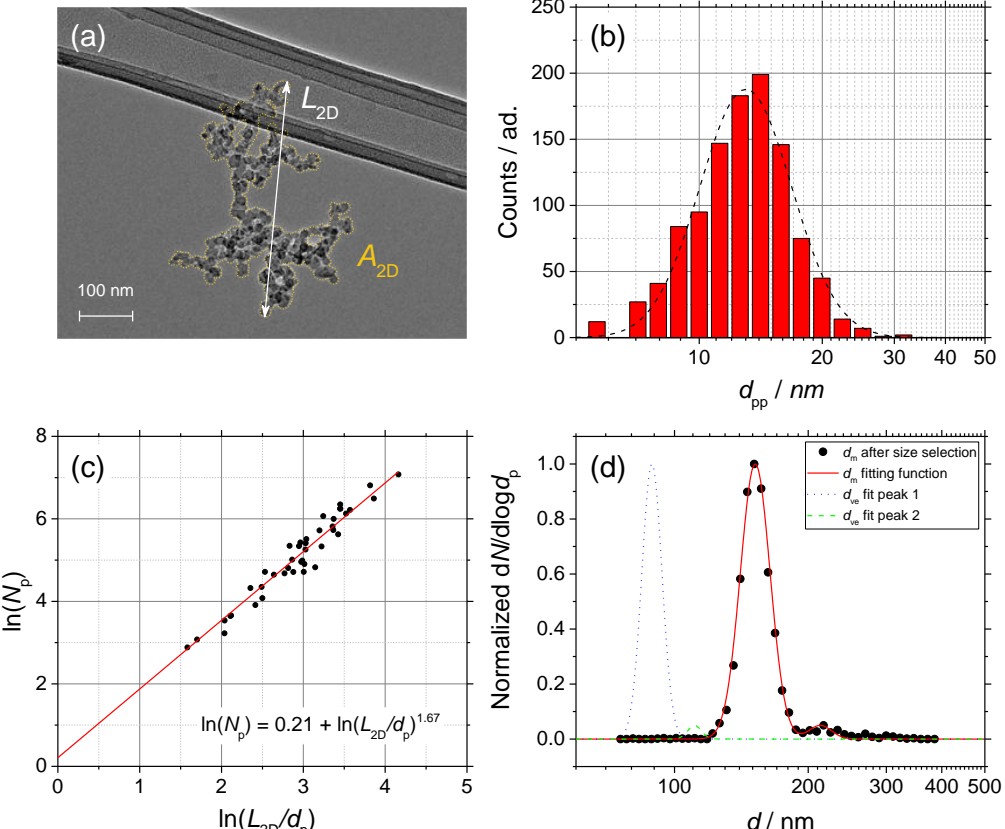

Fig. 5. Soot sampled from the turbulent jet flame supplied with liquid kerosene at 70 mm HAB. (a) TEM picture of a soot aggregate showing $L_{2D}$ and $A_{2D}$. (b) $d_{pp}$ size distribution (red bars) and lognormal fit (black dashed line), $d_{pp}$ = 13.0 nm (mass equivalent $d_{pp}$ = 14.1 nm). (c) $\ln(N_{pp})$ vs. $\ln(L_{2D}/d_{pp})$ plot from which $D_f$ is obtained (42 projections). (d) normalized SMPS data and $d_m$ fit after size selection at 150 nm (black data points and red dashed line), morphology-corrected $d_{ve}$ calculated using Eq. (11) showing a two-mode fit from Eq. (12) (blue dotted and green dashed lines).

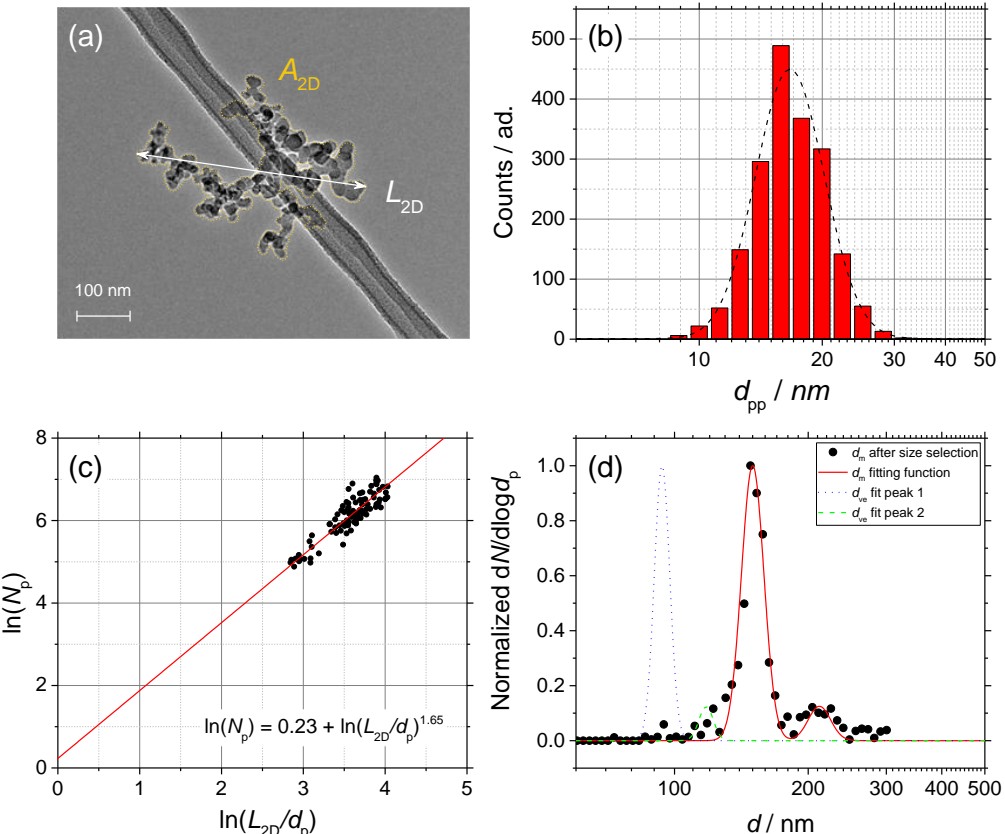

**Fig. 6. Soot sampled from the turbulent jet flame supplied with liquid kerosene at 130 mm HAB. (a) TEM picture of a soot aggregate showing $L_{2D}$ and $A_{2D}$. (b) $d_{pp}$ size distribution (red bars) and lognormal fit (black dashed line), $d_{pp}$ = 16.7 nm (mass equivalent $d_{pp}$ = 17.7 nm). (c) $\ln(N_{pp})$ vs. $\ln(L_{2D}/d_{pp})$ plot from which $D_f$ is obtained (100 projections). (d) normalized SMPS data and $d_m$ fit after size selection at 150 nm (black data points and red dashed line), morphology-corrected $d_{ve}$ calculated using Eq. (11) showing a two-mode fit from Eq. (12) (blue dotted and green dashed lines).**

## 4.3. CCN activity of soot particles

In this section, the data obtained from CCNc activation experiments are used to validate the approach based on the morphology-corrected $d_{ve}$ and to determine $p(\kappa)$. The activation data are reproduced from our past investigation (Grimonprez et al., 2018). Soot is collected from the kerosene jet flame at two HABs to show the different behavior of young and reactive soot (70 mm HAB) and of mature and more inert soot (130 mm HAB) during the chemical aging. The morphology-corrected $d_{ve} = d_{ve}(d_{pp}, D_f, d_m)$ is considered as representative of both the soot particle size distribution ($d_m$) and morphology ($d_{pp}$ and $D_f$) and used as $d_p$ in Eq. (3). $p(\kappa)$ is obtained from the fitting of $F_a = F_a(SS)$ with Eq. (3) and Eq. (4) with $\mu_{\kappa,mode}$ and $\sigma_{\kappa,geo}$ treated as free parameters.

Fig. 7 shows the activation data obtained after chemical aging with ozone and used to quantify $p(\kappa)$ at different ozone exposure (data points). The activation curves (lines) are calculated in four different scenarios, with each individual plot representing one unique combination of $d_m$ or $d_{ve}$ with $\kappa$ or $p(\kappa)$: to represent the particle size distribution, the mobility diameter after size selection (left column) or alternatively the morphology-corrected $d_{ve}$ (right column) is used. To represent the particle hygroscopic activity, a single $\kappa$ value (top row) or a probability distribution $p(\kappa)$ (bottom row) is used in the fitting function.

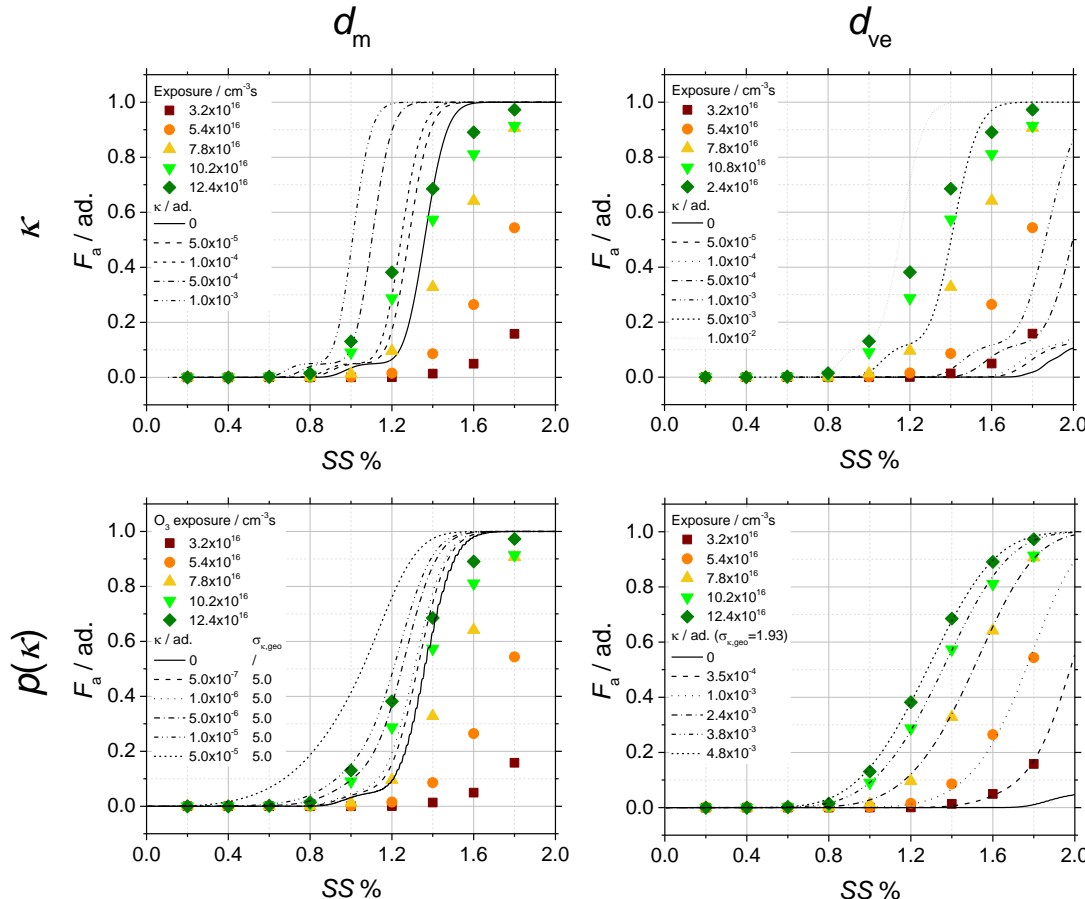

**Fig. 7. Soot sampled from the turbulent jet flame supplied with liquid kerosene at 70 mm HAB. CCNc activation curves of the soot particles obtained after chemical aging with ozone, comparison of the experimental data (data points) to activation curves calculated under four different hypotheses (lines), each of them corresponding to the crossing of one row and one column: single $\kappa$ value (top row), $p(\kappa)$ (bottom row), mobility diameter after size selection (left column), and morphology-corrected $d_{ve}$ (right column).**

As shown in the figure, the calculations based on $d_{m}$ consistently result in the shift of the activation curves to low $SS$, and as a consequence many activation data are found below the Kelvin limit ($\kappa = 0$, black solid line). Although using $p(\kappa)$ instead of $\kappa$ has a clear effect on the slope of the activation curves as $\sigma_{\kappa,geo}$ introduces one additional degree of freedom in the fitting function, this effect is not large enough to compensate for the shift. On the other hand, the calculations based on $d_{ve}$ result in a significant shift of the Kelvin limit to high $SS$. The additional degree of freedom of the fitting function obtained by using $p(\kappa)$ instead of $\kappa$ further improves the quality of the fitting, and the activation data can now be very convincingly reproduced by the calculated curves. In conclusion, in order to correctly reproduce the activation behavior of soot particles, their fractal-like morphology must be taken into account, and a morphology-corrected $d_{ve}$ is a relatively simple and convenient approach. Details on the choice of $\sigma_{\kappa,geo}$ are given in Annex A.

The activation data obtained after chemical aging of mature soot at different ozone exposure (data points) are shown in Fig. 8. Unlike the previous case, the activation curves (lines) are directly calculated using the morphology-corrected $d_{ve}$ and a probability distribution $p(\kappa)$. As shown in the figure, the chemical aging with ozone of mature soot particles with exposure below $\sim 10^{15}$ cm$^{-3}$s produces particles that are only activated at very high $SS$, and the plateau of the activation curves is reached outside the dynamic range of the CCNc. Although it is evident that a weak but consistent activation occurs at high $SS$, this behavior is poorly reproduced

by Eq. (3) and cannot be distinguished from the limit case of ideal non-interacting water vapor and soot particles.

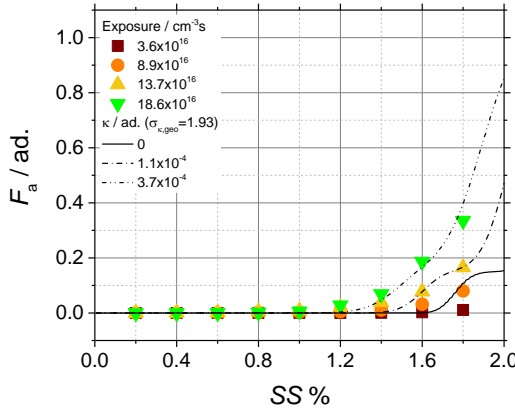

**Fig. 8. Soot sampled from the turbulent jet flame supplied with liquid kerosene at 130 mm HAB. CCNc activation curves of the soot particles obtained after chemical aging with ozone, comparison of the experimental data (data points) to activation curves (lines) calculated by using $p(\kappa)$ the and morphology-corrected $d_{ve}$.**

$\kappa$ of chemically aged soot particles is typically 2-3 orders of magnitude lower than typical inorganic aerosols used in activation experiments (Petters and Kreidenweis, 2007). Therefore, the activation experiments with chemically aged mature soot are particularly interesting as they offer a unique opportunity to estimate the lower limit of validity of Eq. (3). The activation data shown in Fig. 8 are well reproduced by Eq. (3) only for exposure larger than about $13\times10^{-16}$ cm$^{-3}$s and yield $\kappa$ values in the range of $10^{-5}$-$10^{-4}$, while exposure lower than about $9\times10^{-16}$ cm$^{-3}$s result in activation data very close to or below the Kelvin limit. From the comparison of the two situations, it can be estimated that Eq. (3) is only valid for $\kappa > 5\times10^{-6}$.

The data on the size distribution and morphology of the tested young and mature soot particles are summarized and compared to the activation data obtained at different ozone exposure in Table 2.

| HAB / mm | $d_{pp}$ / nm (mode, TEM) | $D_f$ / ad. | $d_m$ / nm (mode, SMPS) | $d_{ve}$ / nm (mode) | Exposure / $10^{16}$ cm$^{-3}$ s | $\mu_{\kappa,geo}$ / $10^{-4}$ (geo. mean) | $\sigma_{\kappa,geo}$ / ad. (geo. st. dev.) |
|---|---|---|---|---|---|---|---|
| 70 | 13.0±0.5 | 1.665±0.005 | 152.1±0.9 | 89.0±0.5 | 3.2 | 5.4 [4.9, 5.6] | 1.93±0.14 |
|  |  |  |  |  | 5.4 | 15 [13, 17] | 1.93±0.14 |
|  |  |  |  |  | 7.8 | 37 [36, 38] | 1.93±0.14 |
|  |  |  |  |  | 10.2 | 59 [59, 60] | 1.93±0.14 |
|  |  |  |  |  | 12.4 | 74 [72, 75] | 1.93±0.14 |
| 130 | 16.7±0.5 | 1.647±0.005 | 150.0±0.9 | 93.8±0.5 | 3.6 | / | / |
|  |  |  |  |  | 8.9 | / | / |
|  |  |  |  |  | 13.7 | 1.7 [1.5, 2.0] | 1.93±0.14 |
|  |  |  |  |  | 18.6 | 5.7 [5.3, 7.3] | 1.93±0.14 |

**Table 2. Size and morphology parameters ($d_{pp}$, $D_f$, $d_m$) used for calculating the morphology-corrected $d_{ve}$ of young (70 mm HAB) and mature (130 mm HAB) soot particles. Comparison with the activation data at different ozone exposure, $\mu_{\kappa,mode}$ and $\sigma_{\kappa,geo}$ are calculated from the best fit (the variability range of $\mu_{\kappa,mode}$ is provided in brackets).**

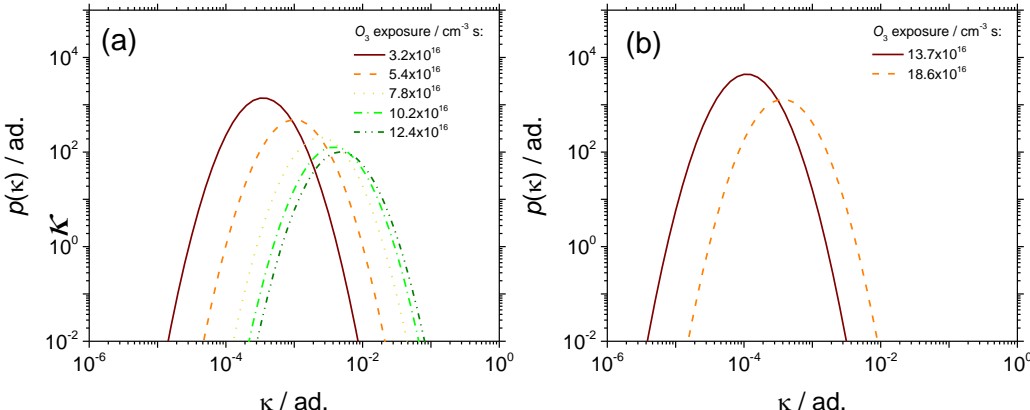

**Fig. 9. Evolution of $p(\kappa)$ vs. ozone exposure of (a) young soot particles sampled at 70 mm HAB, and (b) mature soot particles sampled at 130 mm HAB.**

Fig. 9 shows the effect of the chemical aging on $p(\kappa)$. In the literature, $\kappa$ is often treated as an effective value that folds in all information not explicitly accounted for in the theory, like the change of the water surface tension due to the presence of solutes or the particle morphology for instance. The main advantage of using a morphology-corrected $d_{ve}$ is that the effects of the particle size and morphology on activation are de-coupled from the particle chemistry, and therefore $\kappa$ is preserved as a "chemistry-only" indicator.

As shown in Fig. 9(a), and (b) to a lesser extent, as the ozone exposure increases $p(\kappa)$ shifts to the right ($\mu_{\kappa,\text{mode}}$ increases). The increasing $\mu_{\kappa,\text{mode}}$ is strong evidence that the (surface) chemical composition of the soot particles changes significantly against the ozone exposure time. The PAHs adsorbed at the surface of soot particles are likely candidates to explain the difference in the particle reactivity. Small PAHs typically found adsorbed on the surface of young soot particles have large H/C ratio, and therefore large surface concentration of hydrogen atoms available for abstraction reactions. Conversely, large PAHs typically found on mature soot are characterized by lower H/C ratio that can explain the overall lower reactivity of mature soot particles. An early discussion based on the analysis of $F_a$ at fixed $SS$ can be found in our previous work (Grimonprez et al., 2018).

The comparison of Fig. 9(a) and (b) clearly shows that young and mature soot particles behave very differently to chemical aging and activation experiments. Similar ozone exposure in the range 12 - 14×10⁻¹⁶ cm⁻³s result in $\kappa$ being over two orders of magnitude larger at 70 mm HAB (74×10⁻⁴) than at 130 mm HAB (0.5×10⁻⁴). The soot generation process is therefore critically important for aging/activation experiments and must be taken into account to obtain reliable and reproducible data.

As a concluding remark, it is important to remember that $\kappa$-Köhler theory is a classical theory entirely founded on equilibrium thermodynamics, and for this very reason it is possibly not adapted to describe nanoscale phenomena. The work presented herein remains a simple extension of $\kappa$-Köhler theory that despite not requiring detailed chemical knowledge of the aerosol particles still manages produce rather accurate predictions of the activated fraction of soot particles characterized by complex morphology without adding any ad-hoc hypothesis. A more sophisticated approach is probably required to explain the existence of activation data below the Kelvin limit (adsorption-activation theory for instance). Furthermore, $d_{ve}$ is obviously not the only diameter that can be used to parameterize the particle activation but only a very convenient one. Potentially viable alternatives include the aerodynamic or the gyration diameter for instance, however a systematic verification goes beyond the scope of this paper.

## 5. Conclusions

Soot particles having fractal-like morphology are well known to become, if certain conditions are met, important cloud condensation nuclei (CCN) in the atmosphere. Experimental data is interpreted within the frame of $\kappa$-Köhler theory which allows testing the methodology with soluble (ammonium sulfate) and insoluble or partially soluble (fresh and chemically aged soot) particles, but also comparing the results with other published studies for which $\kappa$ data are available. Thereby, it is possible to introduce in the theory all the parameters needed to describe the size distribution and the complex morphology of soot. In particular, in order to determine the probability distribution $p(\kappa)$ of the hygroscopic parameter $\kappa$ of soot particles, a morphology-corrected volume equivalent diameter $d_{ve}$ is defined and used to parameterize the particle diameter in $\kappa$-Köhler theory. The morphology-corrected $d_{ve}$ is calculated from the particle electrical mobility $d_m$ and folds in two important descriptors of the particle morphology that are the primary particle diameter $d_{pp}$ and the fractal dimension $D_f$. In practice, $d_m$ is measured by scanning mobility particle sizing (SMPS), while $d_{pp}$ and $D_f$ are obtained from the analysis of the projections of soot particles from transmission electron microscopy images (TEM).

This simple model, first tested with isolated, quasi-spherical ammonium sulfate particles, well reproduces the activation experiments and provides $\kappa$ in excellent agreement with the literature. Then, the model is used to determine $p(\kappa)$ of soot particles sampled at different reaction time, or equivalently at different height above the burner (HAB), from a laboratory jet flame supplied with kerosene. To increase their hygroscopic activity, soot particles are chemically aged with ozone beforehand. The experimental activation data are compared to the activation curves calculated using all combinations of $d_m$ or $d_{ve}$ with $\kappa$ or $p(\kappa)$. The best fitting of the activation data is obtained by using $p(\kappa)$ along with the morphology-corrected $d_{ve}$. This important conclusion proves that the particle morphology are shown to plays a non negligible role on the activation of soot particles and has to be taken into account. Furthermore, using the morphology-corrected $d_{ve}$ effectively de-couples the effect of the particle morphology on the activation, potentially preserving $p(\kappa)$ as an indicator of the particle chemistry only.

Young soot particles sampled at 70 mm HAB are more reactive than mature soot particles sampled at 130 mm HAB, resulting in their activation data to be shifted by a significant amount to lower $SS$. The predictive capability of the model is very satisfying in the case of young soot particles that are efficiently converted into CCN after exposure to ozone ($\kappa \sim 3.7 - 74 \times 10^{-4}$). At similar ozone exposure, mature soot particles show a much slower reactivity with ozone ($\kappa \sim 0.5 - 2.7 \times 10^{-4}$) that results in the plateau of the activation curves being reached outside the dynamic range of the CCN counter. This approach proved to be viable for $\kappa > 5 \times 10^{-6}$.

To better take into account the chemical contribution to the process of activation, the next step would be to use a more sophisticated approach by using for instance the adsorption activation theory to describe the first steps of water uptake by fresh and chemically aged soot as a function of relative humidity.

## 6. Annex A. On the role of $\sigma_{\kappa,geo}$ in calculating $F_a = F_a(SS)$

In addition to the geometric mean $\mu_{\kappa,geo}$ of the $p(\kappa)$ distribution, the geometric standard deviation $\sigma_{\kappa,geo}$ of the $p(\kappa)$ distribution is also an important parameter that influences the activation curve $F_a = F_a(SS)$. However, calculations based on Eq. (3) and Eq. (4) are not always sufficiently sensible to account for small changes of $\mu_{\kappa,geo}$ and $\sigma_{\kappa,geo}$ simultaneously. For instance, Fig. 10(a) compares the experimental (black dots) and two calculated activation curves (lines) of ammonium sulfate monodisperse spherical particles. In the figure, two extreme cases have been chosen for the sake of clarity: $\sigma_{\kappa,geo}$ = 1.01 (black dashed line) simulates almost chemically pure aerosol particles, while $\sigma_{\kappa,geo}$ = 1.6 (red solid line) simulates a large variability in the aerosol

particles chemical composition: as clearly shown in the figure, the use of very different $\sigma_{\kappa,geo}$ in the calculations leads to very similar $F_a = F_a(SS)$, as small differences of $\mu_{\kappa,geo}$, that can be easily mistaken for an experimental uncertainly, are able to compensate for a large variability of $\sigma_{\kappa,geo}$.

A sensitivity analysis is performed using the least square function (LSF):

$$\text{LSF} = \sqrt{\frac{\sum_{i=1}^{n}(x_i - y_i)^2}{n-1}} \qquad\qquad \text{Eq. (15)}$$

5 where $x_i$ represent the experimental values of the activated fraction obtained from CCN experiments, and $y_i$ represent the calculated values of the activated fraction with fixed $\mu_{\kappa,geo}$ and free $\sigma_{\kappa,geo}$. As shown in Fig. 10(b), especially if Eq. (3) and Eq. (4) are used to fit experimental data, care has to be taken in the choice of the minimum, as the low variability of the LSF in the range $1.0 < \sigma_{\kappa,geo} < 1.9$ and the shallow local minimum around $\sigma_{\kappa,geo}$ = 1.6 can easily lead to the wrong $\mu_{\kappa,geo}$.

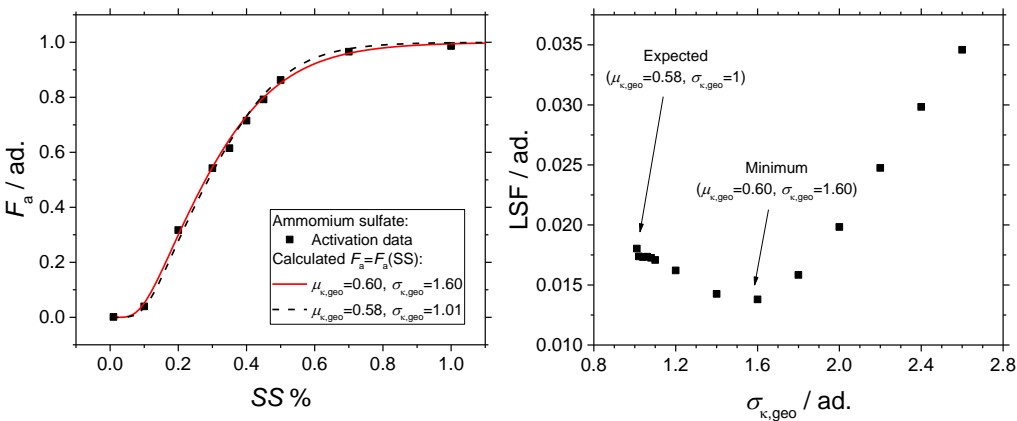

**Fig. 10. Sensitivity analysis performed on ammonium sulfate. (a) comparison of $F_a = F_a(SS)$ calculated from the same aerosol size distribution with $\sigma_{\kappa,geo}$ = 1.01 (black dashed line) or alternatively $\sigma_{\kappa,geo}$ = 1.6 (red solid line). (b) LSF = LSF($\sigma_{\kappa,geo}$).**

Of course, a variability in the chemical composition of the aerosol particles becomes quite complex to treat as
$\sigma_{\kappa,geo}$ > 1, and furthermore is not known *a priori*. In this work, additional calculations have been performed on chemically aged soot. As shown in Fig. 11, the results show that it is not possible to reliably calculate or fit $F_a = F_a(SS)$ by imposing $\sigma_{\kappa,geo} \to 1$, i.e. by assuming a homogeneous chemical composition of the aerosol particles. Therefore, the hypothesis of $\sigma_{\kappa,geo} \to 1$, while holds well for ammonium sulfate, cannot be used for chemically aged soot particles. The stronger impact of $\mu_{\kappa,geo}$ and $\sigma_{\kappa,geo}$ on the calculation of $F_a = F_a(SS)$ on
soot when compared to ammonium sulfate indicates that the fitting process has a physical meaning for chemically aged soot contrarily to its application to pure ammonium sulfate particles. Therefore, for the determination of $\sigma_{\kappa,geo}$, the average value obtained from the fit of the three curves at high ozone exposure (dark green, light green and yellow in Fig. 11) is used that leads to $\sigma_{\kappa,geo}$ = 1.93±0.14. This choice is justified since the three curves clearly reach the plateau at $F_a$ = 1, and hence result to more reliable fits than the two
curves at lower exposure.

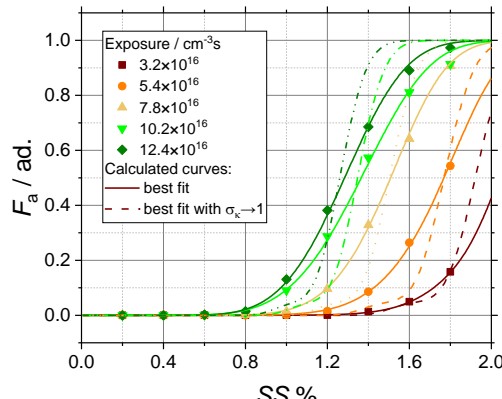

**Fig. 11. Sensitivity analysis performed on chemically aged soot particles sampled from the turbulent jet flame supplied with liquid kerosene at 70 mm HAB. CCNc activation curves of the soot particles obtained after chemical aging with ozone, comparison of the experimental data (data points) to activation curves calculated with $\sigma_{\kappa,geo}$ = 1.01 (dashed lines) and best fit with $\sigma_{\kappa,geo}$ = 1.93±0.14 (solid lines).**

*Data availability.* All data used in this study are available at: https://doi.org/10.4121/uuid:c2f66d57-0e15-43e0-beeb-3120663b7010.

*Author contributions.* JW provided the initial version of the model. JW, SG and SB carried out the research and performed the data analysis. JW, AF, JY, PD and DP improved and refined the model. JW and AF wrote the manuscript with contributions from all authors. All authors have given approval to the final version of the manuscript.

*Competing interests.* The authors declare that they have no conflict of interest.

*Acknowledgments.* This work was supported by the Agence Nationale de la Recherche through the LABEX CAPPA (ANR-11-LABX-0005), the Ministry of Higher Education and Research, Hauts-de-France Regional Council, the European Regional Development Fund (ERDF) through the Contrat de Projets Etat-Region (CPER CLIMIBIO) and the MERMOSE project sponsored by the French Civil Aviation Authority (DGAC).

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
