# Peer review of "Influence of the dry aerosol particle size distribution and morphology on the cloud condensation nuclei activation. An experimental and theoretical investigation"

_Atmospheric Chemistry and Physics, 2019_

## Referee Comment (RC1) · Anonymous Referee #1 · 6 May 2019

Wu et al. present laboratory data and theoretical modeling to treat the influence of particle morphology on CCN activation spectra. Size distribution data and the k-Kohler parameterization are combined to predict the CCN number concentration as a function of supersaturation. Fractal dimension inferred from single particle transmission electron microscopy experiments is folded into the analysis. The method is tested using ammonium sulfate and soot particles from a miniCAST burner. K-values for soot as a function of ozone exposure are presented.

This manuscript is not suitable for publication in ACP, according to the journal standard

to publish studies "with general implications for atmospheric science". The paper does not reach clear new conclusions, mostly focuses on methodology and would be more suitable for AMT. More importantly, the manuscript lacks context of a large body of published literature, needs additional analysis folding in DMA transfer function models, and will require significant rewriting to become acceptable for publication.

Major comments

Eqs. (11) and (12), "the master equations" of the work are simply a cumulative size or supersaturation distributions that have been used in similar for decades in droplet activation schemes or closure studies (e.g. Abdul-Razzak and Gahn, 2000, Snider et al., 2003 and references therein).

In general, the work herein can be interpreted either as a laboratory CCN closure experiment or an improvement to constrain laboratory inferred k-values from data. No studies in that context are cited. For example, it has long been recognized that k-values derived from CCN data are "effective" or "apparent" k-values that have folded in effects of surface tension, solubility, and particle morphology (e.g. Poschl et al., 2009, Sullivan et al., 2010). CCN closure experiments that take size distribution data, composition and shape have been performed extensively on ambient aerosol. Studies that account for particle shape in the calibration of CCN instruments using DMAs and non-spherical particles are routine (e.g. Snider et al., 2006, Rose et al., 2008, Kuwata and Kondo, 2009). Using TEM to obtain particle shape may be novel here (this referee is not 100% sure), but the concepts described in Section 2 are well established.

The "master equation" (Eq. 11) cannot be applied to DMA distributions. The role of the DMA transfer function and of multiply charged particles needs to be taken into account (e.g. Petters, 2018). It is clear that the effect of multiply charge particles is not large as shown by the shoulder in Figure 6. However, the paper makes claims about kappa being a "Dirac delta distribution", i.e. a single value applied to a distribution. The question is whether the observed activation behavior is due to size distribution alone,

or whether shape and composition heterogeneity increases the broadness. Therefore, Eq. (11) should be replaced with the DMA transfer function, including multiply charged particles, and the use unexplained discrepancies to identify the effect of shape and/or heterogeneity.

The influences of size distribution heterogeneity and particle shape on the cumulative CCN spectrum have been discussed in the literature (e.g. Petters et al., 2009, Kuwata and Kondo, 2009, Su et al., 2010, Cerully et al., 2011). The work of Su et al. is particularly pertinent to this paper.

The aging studies of miniCAST soot are interesting. However, it is important to place these results in the context of several similar studies that have investigated the CCN activity of (chemically aged) soot.

The framing around improving k-Kohler or Kohler theory in general is not justified. Kohler theory predicts the activation behavior of a single particle. If one applies that theory recursively to a size distribution with non-uniform composition, or a shaped particle, then those assumption need to be questioned, as it has been in previous studies. Statements that "[f]or all practical purposes, both theories operate under the hypothesis of Dirac delta distributions" don't really make sense in that context.

It is not clear what the main finding of the work is. Equations are presented and somewhat successfully tested against experimental data, but that type of exercise has been presented before. The novelty of the work and the scientific findings contributed need to be better articulated.

The writing of the manuscript needs to be improved significantly. Words are frequently used incorrectly. The writing lacks context of important prior literature. The framing of the work needs to be revised. Section 2 reads more like a tutorial than a scientific paper.

The data availability is inconsistent with the journal policy. The data should be deposited in a publicly available repository.

Other comments

"The modern parametrization of the classical theory of nucleation (k-KÓğhler theory) implicitly assumes a Dirac delta distribution to model the density of ideal spherical point size dry particles and droplets."

To my understanding Kohler theory isn't equivalent to "classical nucleation theory", which usually refers to homogeneous nucleation.

The second part of the sentence isn't quite parsable. Is this talking about density profiles within a particle? Or differences in density and composition within a distribution. Given the context of the paper, it seems to be the latter. k-KÓğhler treats the activation of a single particle. It therefore does not make any assumptions about the heterogeneity of the size distribution.

"To the present day" → please rephrase

"internally mixed nature" → internally mixed refers to a distribution property (i.e. two substances may be internally mixed or externally mixed across a distribution). A single particle is either pure or mixed. k-KÓğhler theory is one means to treat the activation of mixed particles.

"partially soluble" → please change to sparingly soluble. The state of dissolution is determined by the water content. A substance may be "partially dissolved" but it has a strictly defined solubility value.

"neither the classical nor the -KÓğhler theory account for the dry aerosol particle size distribution and morphology" → this is not the objective of those theories. This is a problem on how it is applied in practice.

"more specifically, the crossing of the Kelvin limit has been reported in several instances" → Please explain what is meant by that.

"is the master equation of this work" → this is an odd formulation

"geometric deviation" → geometric standard deviation

"and the fitting is implicitly assumed not to carry any physically meaningful parameter. " → not true. See eg. Cerrully et al. 2011, Figure 3.

"within 20% incertitude" → within 20% uncertainty

"soot emissions are emblematic of human activities" → please reword

"understanding the anthropogenic impact on the atmosphere." → please reword

"understanding the anthropogenic impact on the atmosphere." → please specify what "very representative" means. Is there a metric for this?

"univocally defined" → is defined.

"Young soot"; "mature soot" → Soot aren't biological entities. "Fresh soot" and "chemically aged soot" would appear to be better terms.

"k-Köhler theory is built around the idea that the soluble fraction only affects the heterogeneous nucleation " → this is not strictly true. k-Köhler can be used to parameterize CCN activity descriptively, even if the underlying mechanism is incorrect. Such parameterizations have been termed "apparent k" or "effective k" in the literature."

"k parameterizes all the information on the composition of the droplet approximated as an ideal water solution" → this is incorrect. K is a one parameter activity coefficient model that explicitly models the non-ideality at the point of activation.

"Incidentally, the effect of the dry particle electrical charge on the activation is not considered in this work." → This is unclear what this means. As written, it suggests that particle charge itself affects activation. To my knowledge there is no evidence for such a claim. If it is meant that the effect of particle charge on selected size is not considered, this is correct and needs to be addressed (see major comment).

References

Abdulâ ĂŘRazzak, H., andâ ăGhan, S. J. â ă(â ă2000), â ăA parameterization of aerosol activation: 2. Multiple aerosol types, â ăJ. Geophys. Res., â ă105(â ăD5), â ă6837–â ă6844, doi:10.1029/1999JD901161.

Cerully, K. M., Raatikainen, T., Lance, S., Tkacik, D., Tiitta, P., Petäjä, T., Ehn, M., Kulmala, M., Worsnop, D. R., Laaksonen, A., Smith, J. N., and Nenes, A.: Aerosol hygroscopicity and CCN activation kinetics in a boreal forest environment during the 2007 EUCAARI campaign, Atmos. Chem. Phys., 11, 12369-12386, https://doi.org/10.5194/acp-11-12369-2011, 2011.

Kuwata, M. and Kondo, Y. (2008). Dependence of size-resolved CCN spectra on the mixing state of nonvolatile cores observed in Tokyo. J. Geophys. Res. 113:D19202.

Kuwata, M. and Kondo, Y.: Measurements of particle masses of inorganic salt particles for calibration of cloud condensation nuclei counters, Atmos. Chem. Phys., 9, 5921-5932, https://doi.org/10.5194/acp-9-5921-2009, 2009.

Su, H., Rose, D., Cheng, Y. F., Gunthe, S. S., Massling, A., Stock, M., Wiedensohler, A., Andreae, M. O., and Poschl, U. (2010). Hygroscopicity distribution concept for measurement data analysis and modeling of aerosol particle mixing state with regard to hygroscopic growth and CCN activation. Atmos. Chem. Phys. 10:7489-7503.

Poschl, U., Rose, D., and Andreae, M. O.: Strungmann Forum Report, Clouds in the perturbed climate system: Their relationship to energy balance, atmospheric dynamics, and precipitation. Particle hygroscopicity and cloud condensation nuclei activity, MIT Press, Cambridge, Massachusetts, ISBN 978-0-262-01287- 4, 2009.

Petters, M. D., Carrico, C. M., Kreidenweis, S. M., Prenni, A. J., DeMott, P. J., Collett, J. L., and Moosmüller, H.: Cloud condensation nucleation activity of biomass burning aerosol, J. Geophys. Res.-Atmos., 114, 1–16, https://doi.org/10.1029/2009JD012353, 2009.

[Figure]

Petters, M. D.: A language to simplify computation of differential mobility analyzer response functions, Aerosol Sci. Tech., 52, 1437–1451, https://doi.org/10.1080/02786826.2018.1530724, 2018

Snider, J. R.,ÂăGuibert, S.,ÂăBrenguier, J.‐L., andÂăPutaud, J.‐P.Âă(Âă2003),ÂăAerosol activation in marine stratocumulus clouds: 2. Köhler and parcel theory closure studies,ÂăJ. Geophys. Res.,Âă108, 8629, doi:10.1029/2002JD002692,ÂăD15.

Snider, J.R., M.D. Petters, P. Wechsler, and P.S. Liu,Âă2006:ÂăSupersaturation in the Wyoming CCN Instrument.ÂăJ. Atmos. Oceanic Technol.,Âă23,Âă1323–1339,https://doi.org/10.1175/JTECH1916.1Âă

Sullivan, R. C., Moore, M. J. K., Petters, M. D., Kreidenweis, S. M., Roberts, G. C., and Prather, K. A.: Effect of chemical mixing state on the hygroscopicity and cloud nucleation properties of calcium mineral dust particles, Atmos. Chem. Phys., 9, 3303-3316, https://doi.org/10.5194/acp-9-3303-2009, 2009.

---

## Referee Comment (RC2) · Anonymous Referee #2 · 2 Jul 2019

Wu et al. focus on the CCN activity of particles with irregular shape through a combination of theory and experimental data. Although the study contains interesting material, I feel it cannot be published in its current form and unfortunately must recommend rejection to provide the authors enough time to prepare a modified manuscript.

Overall this study contains a very large amount of material that is not wrong, but is "textbook", often without reference to any of the large body of published literature on the subject. I strongly recommend that the authors give appropriate credit to the published work, but also keep only what is absolutely necessary. Reviewer #1 presents a few

references, I can add also the textbooks of Seinfeld & Pandis, Pruppacher and Klett, as well as review of state-of-the art mechanistic parameterizations (e.g., from the groups of Ghan or Nenes) – which normally adopt formulations that use lognormals (e.g., Fountoukis and Nenes, 2005). There is now a rich literature based upon CCN spectral analysis to constrain hygroscopicity – considering shape factors, multiple charging and DMA transfer function effects (e.g., Cerully et al., 2011 and references therein) – even for well established calibration aerosol like sodium chloride and ammonium sulfate.

What I suggest is that the authors consider a resubmission, focusing on the results of the soot activation experiments, and also comparing them to existing literature. In their analysis it is nice that shape effects are considered, and even nicer that the fractal dimension is explicitly determined, because it allows then to explore other theories of activation such as adsorption-activation theory (e.g., Kumar et al., 2009; Laaksonen et al., 2016 and references therein) – alone or in combination with Kohler theory. It would be interesting to see how this other framework performs and if it can provide insight on the drivers of CCN activity for the particles studied.

The study also suffers from limitations that include the important DMA transfer function effects and multiple charging (although noted on Page 17 by the authors, they do not go through more in-depth analysis), which is considered routine in established groups.

Below some specific comments on aspects of the paper that also should be addressed: Page 2, Line 6: k-Kohler is not just for partially soluble particles, it's to address chemical complexity (mixtures of solutes of a wide range of molar masses) – and can account for partial solubility as well.

Page 2, Line 9: Kohler theory is formulate for a particle. It can (and is) extended to a size distribution easily, as mentioned in the above texts. I don't understand why the authors present this as an issue.

Page 2, Line 10: Sure, you can use - and is even elegant - to use Dirac functions to represent many particles of a given size as a size distribution. I understand where

the authors come from when they say that Kohler theory cannot be applied to a broad size distribution - but I also find it quite narrow because it is equivalent to saying that a size distribution with a finite particle width cannot be described with a monodisperse particle distribution (which is sort of an obvious statement)! On the other hand, Kohler theory can be applied to size distributions, even in the narrowest sense considered by the authors, because it defines the "size" above which all becomes droplets (this of course, assuming the soluble fraction is sufficient, with all that this implies). In this sense, a characteristic size is linked to a characteristic saturation - which is the basis of Kohler theory and the concept of a critical point that comes out of it.

Page 2, Line 10: True, but then you need to use a theory that does not assume at least perfectly wettable particles. The work of Gorbunov, or adsorption activation theory can easily treat such situations – and is physically based. It would be nice if the authors actually refer to that work and consider it in their analysis.

Figures: uncertainties are sometimes included in the activation plots, but they are not propagated later on in any of the analysis. This would be nice to see – and also include uncertainty from the DMA transfer function, shape factor, multiple charges, etc.

Page 19, line 30: I don't understand why nanoparticles will not be in equilibrium. The relevant timescale is extremely small – unless if I misunderstood the point raised by the authors.

References Cerully, K.M., Raatikainen, T., Lance, S., Tkacik, D., Tiitta, P., Petaja, T., Ehn, M., Kulmala, M., Worsnop, D.R., Laaksonen, A., Smith, J.N. and A. Nenes (2011) Aerosol Hygroscopicity and CCN Activation Kinetics in a Boreal Forest Environment during the 2007 EUCAARI Campaign, Atmos.Chem.Phys., 11, 12369-12386

Fountoukis, C. and A. Nenes, A. (2005) Continued Development of a Cloud Droplet Formation Parameterization for Global Climate Models, J.Geoph.Res.,110,D11212, doi:10.1029/2004JD005591

[Figure]

Kumar, P., Sokolik, I.N., and Nenes, A. (2009) Parameterization of Cloud Droplet Formation for Global and Regional models: Including Adsorption Activation from Insoluble CCN., Atmos. Chem. Phys., 9, 2517-2532

Laaksonen, A., Malila, J., Nenes, A., Hung, H.Mop., Chen, J.P. (2016) Surface fractal dimension, water adsorption efficiency, and cloud nucleation activity of insoluble aerosol, Nat.Sci.Rep., 6, 25504, doi:10.1038/srep25504
* * *

---

## Author Response (AR1)

19th August 2019

**Object – Answers to the Reviewers for the paper: "*Influence of the dry aerosol particle size distribution and morphology on the cloud condensation nuclei activation. An experimental and theoretical investigation*", Ref. ACP-2019-172.**

Dear Editor, dear Reviewers,

We would like to thank the Editorial Board for considering our paper «*Influence of the dry aerosol particle size distribution and morphology on the cloud condensation nuclei activation. An experimental and theoretical investigation*» for publication in ACP. We would also like to thank the Reviewers for the extensive amount of work that they clearly dedicated to our paper, we greatly appreciated their constructive comments and suggestions. We have carefully studied the comments, and the original paper has been thoroughly revised accordingly. In particular, as suggested by the Reviewers, the paper goal has been re-framed to focus on the effect of the particle morphology on activation and on the different response to chemical aging of soot having different properties. The data treatment has been updated to account for the multiple charges effect, the DMA transfer function has been used, a probability distribution of the parameter kappa has been introduced and one of the test distributions obtained with ammonium sulfate has been replaced with original results obtained on soot at 70 mm HAB. A detailed point-by-point rebuttal and a marked-up version of the revised manuscript can be found at the end of this document (all modifications are highlighted in blue font color and references to the Reviewers' comments are given in Microsoft Word Revision Mode).

With our best regards,

Dr. Alessandro Faccinetto on behalf of all Authors.

**Answers to anonymous Referee #1.**

**Overview.**

*Wu et al. present laboratory data and theoretical modeling to treat the influence of particle morphology on CCN activation spectra. Size distribution data and the k-Kohler parameterization are combined to predict the CCN number concentration as a function of supersaturation. Fractal dimension inferred from single particle transmission electron microscopy experiments is folded into the analysis. The method is tested using ammonium sulfate and soot particles from a miniCAST burner. K-values for soot as a function of ozone exposure are presented. This manuscript is not suitable for publication in ACP, according to the journal standard to publish studies "with general implications for atmospheric science". The paper does not reach clear new conclusions, mostly focuses on methodology and would be more suitable for AMT. More importantly, the manuscript lacks context of a large body of published literature, needs additional analysis folding in DMA transfer function models, and will require significant rewriting to become acceptable for publication.*

**Major comments.**

*(1) Eqs. (11) and (12), "the master equations" of the work are simply a cumulative size or supersaturation distributions that have been used in similar for decades in droplet activation schemes or closure studies (e.g. Abdul-Razzak and Gahn, 2000, Snider et al., 2003 and references therein).*

Answer. We did our best to cover the most significant works in the field, and we thank the Reviewer for providing some examples of literature that we did not cited or that we missed. It is indeed true that the droplet activation of aerosol particles has been parameterized in the past by using a cumulative function of the lognormal distribution (Abdul-Razzak et al. 1998). The cumulative function has been used as well for predicting the type of the aerosol measured in the outdoor measurements (Snider et al. 2003). However, we would like to notice that especially in recent literature or in works more strictly related to combustion (our field) that investigate the activation of soot particles (Sullivan et al. 2009, Lambe et al. 2015; Tang et al. 2015), sigmoid function are still used for the determination of the critical supersaturation. Therefore, we believed this topic worthy to be discussed. Please notice that in response to this and to comments (2), (3), (4) and (6) the *Introduction*, *Theory* and *Results and Discussion* sections have been heavily updated or entirely rewritten: the more "tutorial-like" parts are reduced to the minimum required for presenting the research context (Eq. 2 in the *Introduction* and section 2.1), all the missing references are added and the paper is re-framed to focus on the effect of the morphology on soot activation.

*(2) In general, the work herein can be interpreted either as a laboratory CCN closure experiment or an improvement to constrain laboratory inferred k-values from data. No studies in that context are cited. For example, it has long been recognized that k-values derived from CCN data are "effective" or "apparent" k-values that have folded in effects of surface tension, solubility, and particle morphology (e.g. Poschl et al., 2009, Sullivan et al., 2010). CCN closure experiments that take size distribution*

*data, composition and shape have been performed extensively on ambient aerosol. Studies that account for particle shape in the calibration of CCN instruments using DMAs and non-spherical particles are routine (e.g. Snider et al., 2006, Rose et al., 2008, Kuwata and Kondo, 2009). Using TEM to obtain particle shape may be novel here (this referee is not 100% sure), but the concepts described in Section 2 are well established.*

Answer. In the original formulation of kappa-Köhler theory (Petters and Kreidenweis 2007), kappa is used to estimate the chemical activity of water in a multicomponent system. Indeed, an effective/apparent kappa has been defined and often used to account for surface tension, solubility and particle morphology effects as mentioned by the Reviewer. Furthermore, a shape factor has been considered to correct the difference between the experimental activation and simulated activation of irregular inorganic aerosol particles (Rose et al. 2008, Snider et al. 2006, Biskos et al. 2006). A similar approach has been used as well on the activation study of soot particles (Lambe et al. 2015), and in particular a volume equivalent diameter has been introduced to avoid the crossing of the Kelvin limit (Tritscher et al. 2011, Lambe et al. 2015). Please also see comment (16) for more details. Following this last approach, in our work we aim to provide a method for the estimation of the volume equivalent diameter $d_{ve}$ of soot particles that accounts for the aggregate size and morphology rather than relegating all the deviations from the expected activation curves to the effective/apparent kappa. In our opinion, "de-coupling" the effective/apparent kappa and integrating size and morphology effects in $d_{ve}$ has the potential advantage of preserving kappa as an indicator of the chemistry at the particle surface for instance. Please notice that in response to this comment, the old section 2 has been completely replaced with the derivation of the morphology-corrected $d_{ve}$.

*(3) The "master equation" (Eq. 11) cannot be applied to DMA distributions. The role of the DMA transfer function and of multiply charged particles needs to be taken into account (e.g. Petters, 2018). It is clear that the effect of multiply charge particles is not large as shown by the shoulder in Figure 6. However, the paper makes claims about kappa being a "Dirac delta distribution", i.e. a single value applied to a distribution. The question is whether the observed activation behavior is due to size distribution alone, or whether shape and composition heterogeneity increases the broadness. Therefore, Eq. (11) should be replaced with the DMA transfer function, including multiply charged particles, and the use unexplained discrepancies to identify the effect of shape and/or heterogeneity.*

*(4) The influences of size distribution heterogeneity and particle shape on the cumulative CCN spectrum have been discussed in the literature (e.g. Petters et al., 2009, Kuwata and Kondo, 2009, Su et al., 2010, Cerully et al., 2011). The work of Su et al. is particularly pertinent to this paper.*

Answer. Being comments (3) and (4) strictly related, one comprehensive answer is provided. We acknowledge that this could have been better explained in our original manuscript, and we thank the Reviewer for providing the pertinent literature references. Indeed, in our original work we used a single value instead of a distribution to describe kappa, and we observed that the geometrical standard deviation ($\sigma_g = 1.08$) of soot distribution recorded by SMPS is smaller than the value ($\sigma_g = 1.16$) in Eq. (22) obtained by fitting experimental activation curves. This discrepancy was acknowledged but not explained. In response to these comments, in the revised version of the paper all calculations have been updated to take into account the DMA transfer function and the effect of multiply charged particles (the effect on the ammonium sulfate is important and cannot be

neglected, however the effect on soot is comparatively small). Furthermore, a distribution rather than a single kappa value was tested. As the Reviewer suggested, the updated $\sigma_g$ values are now consistent, and therefore the kappa distribution (Su et al. 2010, Cerully et al. 2011) has been applied to all calculation and discussed in the revised version of the paper in the updated *Theory* and *Results and Discussion* sections.

*(5) The aging studies of miniCAST soot are interesting. However, it is important to place these results in the context of several similar studies that have investigated the CCN activity of (chemically aged) soot.*

Answer. We would like to emphasize that this work is *not* about miniCAST soot, but about the chemical aging of young and mature soot sampled from a laboratory turbulent jet flame supplied with liquid kerosene. Unlike miniCAST that provides only the soot that can be recovered in the exhausts, laboratory flames allow the generation of soot having controlled properties in terms of aggregate size distribution, morphology and surface chemical composition that are accessed by sampling the flame in the axis at different reaction time (or equivalently height above the burner, HAB). In a miniCAST burner there is no direct access to the flame, and therefore only the exhausts are routinely sampled that leads to several important limitations (contamination of the sampled soot with the exhaust gas phase, sampling of particles formed in very different conditions and mixed only at the end of the combustion process to name a few). Clarifications have been added to the *Introduction*, and the *Results and Discussion* section has been re-framed to focus on this aspect. To the best of our knowledge, the closest studies to our work are (Tritscher et al. 2011) and (Lambe et al. 2015), both referenced in the original paper, in which the approximated equation for kappa > 0.2 is used. The implications on the use of such approximation for the interpretation of the activation curves were discussed in the original paper (old Figure 9), but rendered obsolete after revision. The *Results and Discussion* section has been updated to match.

*(6) The framing around improving k-Kohler or Kohler theory in general is not justified. Kohler theory predicts the activation behavior of a single particle. If one applies that theory recursively to a size distribution with non-uniform composition, or a shaped particle, then those assumption need to be questioned, as it has been in previous studies. Statements that "[f]or all practical purposes, both theories operate under the hypothesis of Dirac delta distributions" don't really make sense in that context.*

Answer. The particle size distribution (Snider et al. 2003) and the heterogeneity of kappa (Su et al. 2010) both influence the activation behavior and have been taken into account in the improved data treatment after revision. The hypothesis of Dirac delta distribution was initially proposed for a group of uniform size particles with constant morphology and chemical composition (for example ammonium sulfate). In our work, the comparison of $\sigma_g$ obtained from experimental SMPS size distribution and from fitted activation spectra demonstrated that the modifications of the heterogeneity of aged soot particles is indeed a concern during the aging process. However, in the revised version of the paper, the main goals have been re-defined to focus on the role of the particle morphology on activation and on the difference in the behavior of young and mature soot. More

emphasis is given to original results rather than well established knowledge. For instance, one distribution on test ammonium sulfate has been replaced with a complete description of the results on soot at 70 mm HAB, and a more detailed discussion has been developed. The *Introduction*, *Theory* and *Results and Discussion* sections have been heavily updated or entirely rewritten. Overall, the paper has been made less pedagogic (removal of the "tutorial" section, focus from the very beginning on the morphology, removed some redundant text).

*(7) It is not clear what the main finding of the work is. Equations are presented and somewhat successfully tested against experimental data, but that type of exercise has been presented before. The novelty of the work and the scientific findings contributed need to be better articulated.*

Answer. All requested modifications are included in the revised version of the paper, please see answers to point (2), (5) and (6) for more details. Briefly, the main results include: "de-coupling" the particle morphology from the effective/apparent kappa and integrating size and morphology effects in $d_{ve}$ to preserving kappa as an "all-chemistry" indicator; providing evidence that chemically aged young and mature soot behave very differently during activation experiments, and thus showing that the soot generation process is critically important for this kind of experiments; providing a lower limit for the validity of the model (kappa$\sim 5\times10^{-6}$) by using chemically aged soot particles characterized by kappa values typically 2-3 orders of magnitude lower than typical inorganic aerosols used in activation experiments. The *Results and Discussion* sections have been heavily updated to focus on the main results.

*(8) The writing of the manuscript needs to be improved significantly. Words are frequently used incorrectly. The writing lacks context of important prior literature. The framing of the work needs to be revised. Section 2 reads more like a tutorial than a scientific paper.*

Answer. The manuscript has been thoroughly revised, and all the corrections suggested by the Reviewers implemented and/or discussed in detail. In particular, the *Introduction* and *Theory* sections have been entirely reworked to remove the feeling of "reading a tutorial" and to include and discuss the missing suggested literature. The *Discussion* section has been modified as well to account for the improved data treatment as suggested. Unfortunately, none of the Authors is a native English speaker. We did our best to keep up with the quality standards of ACP, and kindly thank the Reviewers for all the given suggestions aiming to improve the English level.

*(9) The data availability is inconsistent with the journal policy. The data should be deposited in a publicly available repository.*

Response: All corresponding data will be uploaded in a publicly available repository.

**Other comments.**

*(10) To my understanding Köhler theory isn't equivalent to "classical nucleation theory", which usually refers to homogeneous nucleation.*

Answer. Corrected (this sentence has been removed in the re-framing process).

*(11) The second part of the sentence isn't quite parsable. Is this talking about density profiles within a particle? Or differences in density and composition within a distribution. Given the context of the paper, it seems to be the latter. k-Köhler treats the activation of a single particle. It therefore does not make any assumptions about the heterogeneity of the size distribution."*

Answer. Corrected (this sentence has been removed in the re-framing process).

*(12) To the present day--please rephrase*

Answer. Corrected.

*(13) "internally mixed nature---internally mixed refers to a distribution property (i.e. two substances may be internally mixed or externally mixed across a distribution). A single particle is either pure or mixed. k-Köhler theory is one means to treat the activation of mixed particles."*

Answer. Corrected (this sentence aimed to highlight the mixed nature of the single particle, and it been removed in the re-framing process).

*(14) partially soluble---please change to sparingly soluble. The state of dissolution is determined by the water content. A substance may be "partially dissolved" but it has a strictly defined solubility value.*

Answer. "Partially soluble" is a non-quantitative, descriptive locution commonly used in chemistry to indicate small but unknown solubility product constants $K_{ps}$.

*(15) Response to comment: "neither the classical nor the k-Köhler theory account for the dry aerosol particle size distribution and morphology---this is not the objective of those theories. This is a problem on how it is applied in practice."*

Answer. Corrected (this sentence has been removed in the re-framing process).

*(16) more specifically, the crossing of the Kelvin limit has been reported in several instances---Please explain what is meant by that.*

Answer. According to the kappa-Köhler theory, a non-soluble particle (kappa = 0) could be activated to form a water droplet by increasing the supersaturation up to the so-called Kelvin limit. However, soot particles are sometimes found beyond this limit if the mobility size or the volume equivalent size are used (Tritscher et al. 2011, Lambe et al. 2015). In the frame of kappa-Köhler theory, this leads to k < 0 that has no physical meaning. This point is now clarified in the *Introduction* and *Results and Discussion* sections of the revised paper in the context of the improved data treatment and interpretation suggested by the Reviewer in points (2), (3), (4) and (6).

*(17) is the master equation of this work---this is an odd formulation*

Answer. Corrected (this sentence has been removed in the re-framing process).

*(18) geometric deviation---geometric standard deviation*

Answer. Corrected.

*(19) and the fitting is implicitly assumed not to carry any physically meaningful parameter. --- not true. See Cerrully et al. 2011, Figure 3.*

Answer. A sigmoid curve is used in (Cerully et al. 2011). The slope of the sigmoid curve is explained by the degree of heterogeneity of activated particles. However, if the lognormal distribution of aerosol particles is used, figure 4d in the original paper clearly indicates that the activation curve is far from the sigmoid curve when $\sigma_g$ increases. This point is now clarified in the *Introduction* in the context of the improved data treatment and interpretation suggested by the Reviewer in points (2), (3), (4) and (6).

*(20) within 20% incertitude---within 20% uncertainty*

Answer. Corrected.

*(21) soot emissions are emblematic of human activities---please reword*

Answer. Corrected.

*(22) understanding the anthropogenic impact on the atmosphere---please reword*

Answer. Corrected (this sentence has been removed in the re-framing process).

*(23) please specify what "very representative" means. Is there a metric for this?*

Answer. The fractal dimension obtained for soot particles from 2D TEM analysis in this work spans the range 1.65-1.67 that is quite consistent with values in the range 1.61-1.82 typically found in the literature. Corrected.

*(24) univocally defined---is defined*

Answer. Corrected.

*(25) Response to comment: "Young soot; mature soot---Soot aren't biological entities. Fresh soot and chemically aged soot would appear to be better terms."*

Answer. "Young soot" and "mature soot" are well established descriptive terms used in the combustion community and related to the permanence time of soot inside the flame (or equivalently to the height above the burner, HAB). See for instance (Mitra et al. Combust. Flame 2019, Cain et al. PCCP 2014, Alfé et al. Proc. Combust. Inst. 2009). In a turbulent diffusion flame like the one investigated in this work, "young soot" refers to the condensed phase matter sampled at short reaction time (or low HAB) and characterized by high H/C ratio (typically H/C > 0.7) and small aggregates of 14-15 nm primary particles. On the other hand, "mature soot" refers to the matter sampled at long reaction time (or high HAB), characterized by low H/C ratio (typically < 0.4) that results in a much slower reactivity during the aging process. Furthermore, with respect to young soot, the aggregates are larger both in their number of primary particles per aggregate and in the size of the primary particles (16-17 nm). In other words, it is very well possible to have young fresh soot or young chemically aged soot, opposed to mature fresh soot or mature chemically aged soot. The different behavior of young and mature soot to chemical aging is actually one of the main points of our paper, and such investigation is mainly possible because of the precise control over the flame conditions allowed by the use of a laboratory flame rather than a commercial burner. A more accurate and detailed discussion has been added to the *Introduction*.

*(26) k-Köhler theory is built around the idea that the soluble fraction only affects the heterogeneous nucleation---this is not strictly true. k-Köhler can be used to parameterize CCN activity descriptively, even if the underlying mechanism is incorrect. Such parameterizations have been termed "apparent k" or "effective k" in the literature.*

Response: Corrected (a specific sentence has been added to the *Introduction*).

*(27) k parameterizes all the information on the composition of the droplet approximated as an ideal water solution---this is incorrect. K is a one parameter activity coefficient model that explicitly models the non-ideality at the point of activation.*

Response: Corrected. Please also see answers to point (2).

*(28) Incidentally, the effect of the dry particle electrical charge on the activation is not considered in this work.---This is unclear what this means. As written, it suggests that particle charge itself affects activation. To my knowledge there is no evidence for such a claim. If it is meant that the effect of particle charge on selected size is not considered, this is correct and needs to be addressed (see major comment)."*

Answer. Corrected (the effect of multiply charged particles is now taken into account in the data treatment, and a specific sentence has been added to the *Introduction*).

**Answers to anonymous Referee #2.**

**Overview.**

*Wu et al. focus on the CCN activity of particles with irregular shape through a combination of theory and experimental data. Although the study contains interesting material, I feel it cannot be published in its current form and unfortunately must recommend rejection to provide the authors enough time to prepare a modified manuscript.*

*Overall this study contains a very large amount of material that is not wrong, but is "textbook", often without reference to any of the large body of published literature on the subject. I strongly recommend that the authors give appropriate credit to the published work, but also keep only what is absolutely necessary. Reviewer #1 presents a few references, I can add also the textbooks of Seinfeld & Pandis, Pruppacher and Klett, as well as review of state-of-the art mechanistic parameterizations (e.g., from the groups of Ghan or Nenes) – which normally adopt formulations that use lognormals (e.g., Fountoukis and Nenes, 2005). There is now a rich literature based upon CCN spectral analysis to constrain hygroscopicity – considering shape factors, multiple charging and DMA transfer function effects (e.g., Cerully et al., 2011 and references therein) – even for well established calibration aerosol like sodium chloride and ammonium sulfate.*

**Major comments.**

*(1) What I suggest is that the authors consider a resubmission, focusing on the results of the soot activation experiments, and also comparing them to existing literature. In their analysis it is nice that shape effects are considered, and even nicer that the fractal dimension is explicitly determined, because it allows then to explore other theories of activation such as adsorption-activation theory (e.g., Kumar et al., 2009; Laaksonen et al., 2016 and references therein) – alone or in combination with Kohler theory. It would be interesting to see how this other framework performs and if it can provide insight on the drivers of CCN activity for the particles studied.*

Being comments (1) and (6) strictly related, one comprehensive answer is provided. We would like to thank the Reviewer for the kind and constructive suggestions. A revised version of the paper has been prepared that includes all requested modifications, and indeed focuses on the results of the effect of the morphology on soot particles activation experiments. For a detailed explanation, please see the answers to points (5), (6) and (25) of Reviewer#1. The *Introduction* and *Results and Discussion* sections are particularly affected by the re-framing of the paper. Exploring other theories of activation, unfortunately, is beyond the scope of the present work that instead aims to further develop kappa-köhler theory to account for effects (morphology, and chemistry of soot) not yet fully understood. Nevertheless, we acknowledge this suggestion as an interesting research direction for future developments that we will certainly keep as a perspective for future work.

*(2) The study also suffers from limitations that include the important DMA transfer function effects and multiple charging (although noted on Page 17 by the authors, they do not go through more in-depth analysis), which is considered routine in established groups.*

Answer. We would like to thank the Reviewer for the constructive suggestions and references. Indeed, including $d_{ve}$, multiple charges effects, DMA transfer function and kappa distribution is the proper way to treat the CCN activation data. In the revised version of the paper, the experimental data have been re-treated to account for the above mentioned effects, and a morphology-corrected $d_{ve}$ is then calculated, using the data from TEM measurements. Furthermore, in the revised manuscript we updated the theory, the context of the presentation and all the figures. For more details on specific points, please see the answers to points (1), (2), (3), (4) and (6) of Reviewer#1.

**Specific comments.**

*(3) Page 2, Line 6: k-Köhler is not just for partially soluble particles, it's to address chemical complexity (mixtures of solutes of a wide range of molar masses) – and can account for partial solubility as well.*

Answer. Corrected.

*(4) Page 2, Line 9: Kohler theory is formulate for a particle. It can (and is) extended to a size distribution easily, as mentioned in the above texts. I don't understand why the authors present this as an issue.*

Answer: We acknowledge that this could have been explained in a different and more appropriate way, and we thank the Reviewer for providing the pertinent literature references. Please notice that in response to this comment and to comments (1), (2), (3) and (4) of Referee #1, the *Introduction* and *Theory* sections have been entirely rewritten.

*(5) Page 2, Line 10: Sure, you can use - and is even elegant - to use Dirac functions to represent many particles of a given size as a size distribution. I understand where the authors come from when they*

*say that Kohler theory cannot be applied to a broad size distribution - but I also find it quite narrow because it is equivalent to saying that a size distribution with a finite particle width cannot be described with a monodisperse particle distribution (which is sort of an obvious statement)! On the other hand, Kohler theory can be applied to size distributions, even in the narrowest sense considered by the authors, because it defines the "size" above which all becomes droplets (this of course, assuming the soluble fraction is sufficient, with all that this implies). In this sense, a characteristic size is linked to a characteristic saturation - which is the basis of Kohler theory and the concept of a critical point that comes out of it.*

Answer. We thank the Reviewer for this correction, in the revised version of the paper we updated the *Introduction* and *Theory* sections accordingly. Please also see answers to comments (3) and (4) of Reviewer#1.

*(6) Page 2, Line 10: True, but then you need to use a theory that does not assume at least perfectly wettable particles. The work of Gorbunov, or adsorption activation theory can easily treat such situations – and is physically based. It would be nice if the authors actually refer to that work and consider it in their analysis.*

Answer. Please see answer to comment (1).

*(7) Figures: uncertainties are sometimes included in the activation plots, but they are not propagated later on in any of the analysis. This would be nice to see – and also include uncertainty from the DMA transfer function, shape factor, multiple charges, etc.*

Answer. All missing uncertainties have been added to plots and tables.

*(8) Page 19, line 30: I don't understand why nanoparticles will not be in equilibrium. The relevant timescale is extremely small – unless if I misunderstood the point raised by the authors.*

Answer. Corrected. This sentence was not clear, and it has been removed.

[revised manuscript text omitted]

**2. Theory**

**2.1. Modification of $F_{\mathrm{a}}(SS)$ to include a distributions of $d_{\mathrm{p}}$ and $\kappa$**

Probability density functions of $d_{\mathrm{p}}$ (Abdul-Razzak and Ghan, 2000; Snider et al., 2006) and $\kappa$ (Cerully et al., 2011; Su et al., 2010) have been widely used in aerosol science and atmospheric research to describe the CCN

**Comment [S6]:** Reviewer#1 (16).

**Comment [S7]:** Reviewer#1 (19).

**Comment [S8]:** Reviewer#1 (26).

**Comment [S9]:** Reviewer#1 (1), (3) and (4). Reviewer#2 (2), (4) and (5).

The content of this section now very briefly reviews the "practical approach" to fit the activated fraction curves that considers a distribution of $\kappa$ values instead of a single value.

activity. By using lognormal distributions, and by treating $d_{\mathrm{p}}$ and $\kappa$ as uncorrelated variables to avoid double integration (Su et al., 2010; Zhao et al., 2015), the activated fraction $F_{\mathrm{a}}(SS)$ can be calculated as:

$$F_{\mathrm{a}}(SS) = \sum_{\kappa=0}^{\infty} \left\{ \frac{1}{2} - \frac{1}{2}\operatorname{erf}\left[ \frac{\ln d_{\mathrm{p}}(\kappa, SS) - \ln \mu_{\mathrm{p,mode}} - \ln^2 \sigma_{\mathrm{p,geo}}}{\sqrt{2}\ln \sigma_{\mathrm{p,geo}}} \right] \right\} p(\kappa)\Delta\kappa \qquad \text{Eq. (3)}$$

where $\mu_{\mathrm{p,mode}}$ and $\sigma_{\mathrm{p,geo}}$ are the mode and the geometrical standard deviation of $d_{\mathrm{p}}$. $p(\kappa)$ is the probability density function of $\kappa$:

$$p(\kappa) = \frac{1}{\kappa \ln \sigma_{\mathrm{\kappa,geo}} \sqrt{2\pi}} e^{-\frac{\left[\ln \kappa - \ln \mu_{\mathrm{\kappa,mode}} - \ln^2 \sigma_{\mathrm{\kappa,geo}}\right]^2}{2\ln^2 \sigma_{\mathrm{\kappa,geo}}}} \qquad \text{Eq. (4)}$$

5  where $\mu_{\mathrm{\kappa,mode}}$ and $\sigma_{\mathrm{\kappa,geo}}$ are the mode and the geometrical standard deviation of $\kappa$.

**2.2. Definition of the morphology-corrected volume equivalent diameter $d_{\mathrm{ve}}$**

In this section, we find an original relationship to derive $d_{\mathrm{ve}}$ from $d_{\mathrm{m}}$ for a fractal-like aggregate. $d_{\mathrm{ve}}$ is the diameter of a sphere having the same volume as the aggregate, and assuming the aggregate made of identical, spherical primary particles, is defined as:

$$d_{\mathrm{ve}} = d_{\mathrm{pp}} N_{\mathrm{pp}}^{\frac{1}{3}} \qquad \text{Eq. (5)}$$

10  where $d_{\mathrm{pp}}$ and $N_{\mathrm{pp}}$ are the diameter and number of primary particles per aggregate, respectively. It is worth to notice that often, for practical purposes, the value of $d_{\mathrm{pp}}$ used in calculations is the mass equivalent diameter of the primary particle distribution obtained from TEM measurements. On the other hand, $d_{\mathrm{m}}$ is directly linked to the aerodynamic force acting on the particle $F_{\mathrm{drag}}$ (Dahneke, 1973; Tritscher et al., 2011) and can be directly obtained from SMPS measurements:

$$F_{\mathrm{drag}} = \frac{3\pi\eta d_{\mathrm{m}} v_{\mathrm{r}}}{C_{\mathrm{c}}(d_{\mathrm{m}})} \qquad \text{Eq. (6)}$$

15  where $\eta$ and $v_{\mathrm{r}}$ are the kinematic viscosity of the gas and the particle-gas relative velocity, and $C_{\mathrm{c}}$ is the Cunningham slip factor (Allen and Raabe, 1985):

$$C_{\mathrm{c}}(K_{\mathrm{n}}) = 1 + K_{\mathrm{n}}\left[1.142 + 0.558 \exp\left(-\frac{0.999}{K_{\mathrm{n}}}\right)\right] \qquad \text{Eq. (7)}$$

$K_{\mathrm{n}} = 2\lambda_{\mathrm{g}}/d_{\mathrm{m}}$ is the Knudsen number and $\lambda_{\mathrm{g}}$ is the gas mean free path. The drag force acting on an aggregate $F_{\mathrm{drag,agg}}$ can be approximated using the drag force acting on each primary particle $F_{\mathrm{drag,pp}}$, which is considered as a sphere, using the relation (Yon et al., 2015):

$$F_{\mathrm{drag,agg}} = F_{\mathrm{drag,pp}} N_{\mathrm{pp}}^{\frac{\Gamma}{D_{\mathrm{f}}}} \qquad \text{Eq. (8)}$$

20  The exponential factor $\Gamma = \Gamma(d_{\mathrm{pp}})$ has been empirically estimated as a function of the Knudsen number (Yon et al., 2015) for soot particles generated with a miniCAST commercial burner (propane-air diffusion flame). In the range 1.61 < $D_{\mathrm{f}}$ < 1.79:

**Comment [S10]:** Reviewer#1 (2) and (6). Reviewer#2 (1).

The content of this section corresponds to the section 4.2 of the original paper (mostly unchanged, except for minor English corrections). It has been moved here in compliance to the demands of Reviewer#1 of making the paper less pedagogic (2) and re-framing the main topic (6) to focus on the effect of the aggregate morphology on activation.

$$\Gamma = 1.378 \left[\frac{1}{2} + \frac{1}{2}\mathrm{erf}\left(\frac{K_n(d_{pp}) + 4.454}{10.628}\right)\right]$$

Eq. (9)

Although the variability range of $D_f$ might seem quite restrictive, in practice it covers a region representative of soot aggregates (Kelesidis et al., 2017; Yon et al., 2015). Therefore, we make the additional hypothesis that Eq. (9) can be applied to a variety of experimental investigations including our case. Introducing Eq. (6) in Eq. (8) yields:

**Comment [S11]:** Reviewer#1 (23).

**Comment [S12]:** Reviewer#1 (6). Reviewer#2 (1).

This section (old section 4.2.3) has been integrated here as part of the re-framing of the paper.

[revised manuscript text omitted]
_{\text{p}})}{\mathrm{d}d_{\text{p}}} = \sum_{i=1}^{2} \frac{N_i}{d_{\text{p}} \ln \sigma_{\text{p,geo,i}} \sqrt{2\pi}} e^{-\frac{\left[\ln d_{\text{p}} - \ln \mu_{\text{p,mode,i}} - \ln^2 \sigma_{\text{p,geo,i}}\right]^2}{2 \ln^2 \sigma_{\text{p,geo,i}}}} \qquad \text{Eq. (12)}$$

For isolated, spherical and homogeneous particles, $d_{\text{ve}} = d_{\text{m}} = d_{\text{p}}$ (Eggersdorfer and Pratsinis, 2014; Sorensen, 2011). Although Fig. 3 shows a particularly favorable case, in this work it is found that $d_{\text{m}} = d_{\text{p}}$ is always true within 20% uncertainty. A summary of the parameters of the distributions is given in Table 1.

**Comment [S13]:** Reviewer#1 (3), (4) and (28). Reviewer#2 (2).

Multiple charges and the DMA transfer function are now considered in the calculations.

[revised manuscript text omitted]

**Comment [S18]:** Please notice that the *Conclusions* have been updated to match the modifications required by the Reviewers.

[revised manuscript text omitted]

---

## Author Response (AR2)

**Object – 2$^{nd}$ review and answers to the Reviewers for the paper: "*Influence of the dry aerosol particle size distribution and morphology on the cloud condensation nuclei activation. An experimental and theoretical investigation*", Ref. ACP-2019-172.**

Dear Editor, dear Reviewers,

We have carefully studied the Reviewers' comments and further improved the paper accordingly. We believe that our approach is now correct (based on Su et al. and Cerully et al. papers as suggested by the Reviewers in the first revision), and that the inconsistent results on ammonium sulfate activation were due to the lack of a numerical analysis for the correct determination of the best fitting function for $F_a$. In this 2$^{nd}$ revision, a sensibility analysis is proposed to explain the discrepancy between the expected and calculated width of the kappa distribution of ammonium sulfate, and the discussion is extended to the chemically aged soot (brief discussion in the main text, main discussion can be found in the new Annex A). A discussion on the adsorption activation theory is also included in this revised version of the paper. A detailed point-by-point rebuttal and a marked-up version of the revised manuscript can be found at the end of this document (all modifications and references to the Reviewers' comments are highlighted in Microsoft Word Revision Mode).

Regardless of the final decision of the Editorial Board, once again we would like to thank the Editorial Board and especially the Reviewers for their patience and the extensive amount of work they dedicated to our paper. We greatly appreciated their constructive comments and suggestions that allowed us to progress much in the data analysis and interpretation of the results.

With our best regards,

Dr. Alessandro Faccinetto on behalf of all Authors.

**Answers to anonymous Referee #1.**

**Major comments**

*(1) The authors refocused the work and added citations about previous work to the manuscript given by both referees. However, the authors failed to properly implement those previous works in their analysis. Important experimental details are not given. As a result, the current way the results are analyzed remains insufficient to warrant publication. Specifically, the authors fail to properly account for DMA transfer effects. For example, Figure 4 shows a kappa frequency distribution for ammonium sulfate. If the effects of non-monodisperse size (not even accounting for multiple charges) had been taken into account properly, the kappa distribution should approach a Dirac delta function. It does not, and thus viewed from a high-level, the result is simply incorrect. The conclusion that "The main advantage of using a morphology-corrected is that the effects of the particle size and morphology on activation are decoupled from the particle chemistry, and therefore is preserved as a "chemistry-only" indicator." is thus not true.*

Answer. In the 1$^{st}$ revision of the paper, the DMA transfer function and the effect of multiple charges were already taken into account following the suggestions of both Reviewers. The inversion found that an excellent approximation of the kappa distribution could be provided by a lognormal distribution with $\mu_{\kappa,geo}$ = 0.60 and $\sigma_{\kappa,geo}$ = 1.6 (Levenberg–Marquardt algorithm to solve the least squares regression, 10$^{-9}$ $\chi^2$ tolerance). At first, we ourselves were puzzled by this result, since we agree with the Reviewer that the kappa distribution of ammonium sulfate (Fig. 4b), once size and multiple charges are taken into account, should tend to a delta function. However, in the 1$^{st}$ revision of the paper and *after* taking into account the DMA transfer function and the effect of the multiple charges, we did not immediately realize that the model is not sensible enough to account for small changes of the position $\mu_{\kappa,geo}$ and width $\sigma_{\kappa,geo}$ of the kappa function simultaneously for monodisperse spherical particles (an example can be found in the figure below, left panel). In this 2$^{nd}$ revision of the paper, after carefully re-analyzing all results, we came to the conclusion that this was caused by a lack of sensibility of the inversion process based on the fitting procedure. In consequence, we performed a detailed sensibility analysis using the least square function (LSF) to estimate the quality of the fitting:

$$\text{LSF} = \sqrt{\frac{\sum_{i=1}^{n}(x_i - y_i)^2}{n-1}}$$

where $x_i$ represent the experimental values of the activated fraction obtained from CCN experiments, and $y_i$ represent the calculated values of the activated fraction with fixed $\mu_{\kappa,geo}$ and free $\sigma_{\kappa,geo}$. LSF = LSF($\sigma_{\kappa,geo}$) is shown in the figure below (right panel): the local minimum near $\sigma_{\kappa,geo}$ = 1.6 was initially chosen as the best fit. However, in the range 1.0 < $\sigma_{\kappa,geo}$ < 1.9 the variability of LSF = LSF($\sigma_{\kappa,geo}$) is small, and therefore the fit returns similar values of $\mu_{\kappa,geo}$ regardless of $\sigma_{\kappa,geo}$ (0.58 < $\mu_{\kappa,geo}$ < 0.60, consistent with the literature within 5% uncertainty). For this reason, in this 2$^{nd}$ revision of the paper, we impose $\sigma_{\kappa,geo} \rightarrow 1$ as a fitting condition: this approach is now explicitly discussed in the text, Annex A has been added to discuss the fitting function quality, Fig. 4 has been

modified by removing Fig. 4b and the corresponding value of $\mu_{\kappa,geo}$ = 0.58 obtained with $\sigma_{\kappa,geo} \to 1$ is given instead.

[Figure]

Of course, this raises the question about the application of the same analysis for soot particles even if the morphology is integrated in the model, since $\sigma_{\kappa,geo}$ > 1 is expected in the general case. Consequently, additional calculations have been performed for chemically aged soot as shown in the figure below. These results highlight that it is not possible to reliably calculate/fit $F_a$ curves for soot particles by imposing $\sigma_{\kappa,geo} \to 1$, i.e. by assuming a homogeneous chemical composition of the chemically aged soot particles. Therefore, the hypothesis of $\sigma_{\kappa,geo} \to 1$, while holds well for ammonium sulfate, cannot be used for chemically aged soot particles. Also, the sensitivity analysis shows a stronger impact of $\mu_{\kappa,geo}$ and $\sigma_{\kappa,geo}$ on the $F_a$ curves for soot compared to ammonium sulfates, indicating that the fitting process is much more robust and provides a physical meaning for soot contrarily to its application to spherical ammonium sulfate particles.

[Figure]

Therefore, in this 2$^{nd}$ revision of the paper, we also revised the calculations of the kappa distribution of chemically aged soot to more accurately account for the effect of $\sigma_{\kappa,geo}$. In particular, we used as $\sigma_{\kappa,geo}$ the average value obtained from the fit of the three curves at high exposure (dark green, light green and yellow in the figure) that clearly reach the plateau at $F_a$ = 1 ($\sigma_{\kappa,geo}$ = 1.93±0.14), and hence result to a more reliable fit than the two curves at lower exposure.

In conclusion, to evaluate kappa the important required information is the mobility distribution function injected in the CCNc. As demonstrated by the comparison with the TEM images in Fig. 3(b) and (c), the impact of the DMA transfer function is minor and overshadowed by the lack of sensibility of the model to $\sigma_{\kappa,geo}$.

*(2) The manuscript is lacking proper treatment of the experimental details. For example, it is unclear where neutralizers were used and where not. Multiple charges are fitted through Eq. (12) but DMA theory relating to charging efficiencies and sizes is not included. The attribution to +1 and +2 contributions shown in Figure 3c are incorrect.*

Answer. The distribution shown in Fig. 1 is the mobility size distribution of the soot particles inside the reactor chamber measured by a second SMPS in tandem (both neutralizers on). Said distribution also provides a good approximation of the mobility size distribution of the soot particles injected in the CCNc, whence the confusion in the previous version of the paper. It can be clearly seen that the aerosol is not perfectly monodisperse ($\sigma_{\kappa,geo} \approx 1.06\text{-}1.10$). Inevitably, after passing through the DMA, some multiply charged particles are present and have to be considered since they contribute to the droplet nucleation, and thus affect the CCN measurements. This is clearly visible for soot particles, as indicated by the presence of a second small peak around 210 nm attributed to particles charged +2. That distribution is determined by using a second neutralizer and DMA in tandem to the first ones. For these distributions, the charge correction is considered in order to have the best quantification of the presence of multiple charges particles entering the CCNc. Information on the use of the neutralizers has been added in the text, and Fig. 3c has been corrected.

*(3) I continue to believe that kappa distributions (or shape factor distributions) from these data could be obtained. Closure of DMA/SMPS/CCN derived shape distributions with TEM would be valuable contribution to the literature. However this requires precise treatment of the tandem DMA and CCN measurements that removes the ambiguity of size, which requires appropriate non-trivial inversion of the data. Successful inversion should remove the effect of the DMA transfer function and retrieve the true hygroscopicity distribution, which in case of the soot, might be all attributed to shape or to oxidation state. Details about neutralizer placement and DMA transfer models should be discussed. The quality of the inversion should be assessed perhaps for a case with synthetic data. Methodology should all be discussed prior to results.*

Answer. Please see the answer to major comment (1) and (2). In response to the request of discussing all methodology before results, the use of Eq. (12) is now discussed at the end of the Theory section in the dedicated subsection 2.3.

**Other comments**

(4) *"To avoid this problem, kappa has been obtained from the fitting of the activation curve with generic sigmoid functions that do not take into account the particle size distribution or a distribution of values of kappa. More specifically, kappa has been calculated from the critical supersaturation (F = 0.5) by using an analytical approximation of Eq. (2) only valid for k > 0.2, which is not always the case for soot particles."* *Who did this for what type of aerosol? Please provide citations to studies that wrongly calculated kappa for soot particles. Even if this were true the above seems to not fully capture what has been done in the literature.*

Answer. The second part of this sentence has been removed.

(5) *ammonium sulfate nanoparticles dispersed in nitrogen → did you account for viscosity differences of using pure N2 and air?*

Answer. The difference in kinematic viscosity at 20°C and 1 bar between air ($15.06\times10^{-6}$ $m^2$ $s^{-1}$) and nitrogen ($15.27\times10^{-6}$ $m^2$ $s^{-1}$) is around 1%, which is over one order of magnitude smaller than other sources of experimental uncertainty, and was thus considered as negligible.

(6) *What is the justification for Eq. (4)? Why would kappa take on this distribution? Should an inversion not retrieve the distribution without prior specification?*

Answer. The idea of using lognormal distributions of kappa comes from Su et al., Atmos. Chem. Phys. 10 (2010) 7489-7503. The authors propose the hypothesis that a lognormal distribution of kappa can lead to a better fit of the CCN data, and demonstrate its applicability on two different atmospheric aerosols. They conclude that: "*Lognormal distribution functions were found to be suitable for approximately describing the hygroscopicity distribution of aerosols in polluted megacity air as well as in pristine rainforest air as determined by size resolved CCN measurements*". The text in the Theory section was modified to make the reference to Su et al. more clear.

(7) *Figure 1 shows a DMA transfer function, but the specifics have not been discussed. The manuscript should be organized in a more sequential manner.*

Answer. Fig. 1 shows the electrical mobility distribution of soot particles measured inside the reactor chamber. This distribution is obtained by SMPS with the neutralizer on, and it is considered as a good approximation of the aerosol size distribution subsequently injected into the CCNc. We recognize that our choice of words might induce some confusion between "mobility distribution of soot particles injected into the CCNc" and "DMA transfer function" that were used interchangeably. This is now corrected, and the whole paper has been revised to remove this ambiguity.

*(8) Figure 1 x-axis should be Dve?*

Answer. Fig. 1 compares the electrical mobility $d_m$ after size selection (black solid line) to the calculated volume equivalent diameter distributions $d_{ve}$ (colored dashed lines), therefore a more generic "$d$" was preferred to avoid confusion, while the details are given in the legend.

*(9) Section 3.1: what is the purity of the water?*

Answer. This information has been added in the methodology section.

*(10) pg. 8: extremely low kappa → what does that mean? Provide a number and reference?*

Answer. This sentence has been rephrased.

*(11) so called → please avoid this phrasing*

Answer. Corrected.

*(12) pg.9 please provide sheath to sample flow ratios for all DMAs. Also, please provide the aerosol neutralization method and activity of sources.*

Answer. Information on the sheath to sample flow ratio was already provided in section 3.4. Information on the neutralization method has been added in the same short section dedicated to the description of the SMPS apparatus, and also added in the text.

*(13) SMSP at regular time intervals → SMPS?*

Answer. Corrected.

*(14) Section 3.4 Diagnostics: "The DMA can be used independently to select aerosol particles of the desired mobility." → was the SMPS DMA ever used this way? If not, please omit as it is confusing.*

Answer. This sentence has been removed for sake of clarity.

*(15) pg. 10, Eq. (12), and Figures 3 + 4:*

*Eq. (12) appears to be an empirical two mode fit that is applied to the observed mobility distribution. There are numerous questions that need to be addressed.*

*(a) Was a neutralizer used in the SMPS? Or were the particles sampled without reneutralization from the chamber?*

Answer. The particles were re-neutralized when sampled from the chamber.

*(b) Was the size distribution shown in Figure 2 inverted? If so, what is the justification for the two charge fit? If not, what is the justification to work with raw spectra?*

*(c) The number concentration and size of the +2 charged particles should not be allowed to be "floating". The number concentration of +2 charged particles is determined strictly by the charging efficiency and that is known a priori.*

Answer. Being points *(b)* and *(c)* strictly related, one comprehensive answer is provided. The SMPS scan approximates well the size distribution in the chamber before the aerosol injection in the CCNc. The empirical fit using two lognormal functions is simply a convenient approach that allows maintaining the analytical approach. The text has been corrected accordingly, and a brief discussion is now provided in the new section 2.3.

*(d) The kappa distribution shown in Figure 4 cannot possibly be correct. All ammonium sulfate particles have the same kappa. This means that the shown kappa distribution does not remove the effect of the DMA transfer function. What I shown here appears to be the kappa one would retrieve not properly accounting for the distribution of sizes produced by the DMA. Such a kappa distribution is not particularly meaningful as it folds in the effect of particle size.*

Answer. We agree with the Reviewer that the kappa distribution of ammonium sulfate should tend to a delta function. At the same time, the effect of the particle size and charge distribution and the DMA transfer function were taken into account in the calculations as suggested by the Reviewers in the first revision of the paper. As shown in this 2[nd] revision of the paper, this unexpected effect could rather be attributed to a lack of sensibility of the fitting function to both the width of the particle distribution and to their kappa at the same time. Please see the answer to major comment (1).

*(e) The authors should discuss observed shape effects for AS in the literature (various sources provide a dynamic shape factor for AS particles).*

Answer. One of the central points of the paper is to introduce a morphology-corrected equivalent diameter $d_{ve}$ for taking into account in the same variable both the size distribution and the morphology of fractal-like aerosols. This is an alternative approach to the shape factor that represents much better the soot morphology as it relies on a tested semi-empirical model (Yon et al., 2015). In this frame, as explained at the beginning of Section 4, for isolated, spherical and homogeneous particles, $d_{ve} = d_m = d_p$. Therefore, in our opinion, it would be incorrect to directly

compare data obtained from ammonium sulfate corrected by the shape factor to data obtained from soot corrected by Yon's model.

*(16) While kappa has it's application, perhaps even for soot, the authors should at least discuss the various adsorption theories that have been proposed (e.g. Henson, 2007 JGR, Sorjamaa and Laaksonen, 2007, ACP, Kumar et al., 2009, ACP), and how those approach differ from the ones used here.*

Answer. Please see answer to Reviewer#2, (1).

**Answers to anonymous Referee #2.**

**Major comments.**

*Thank you for carefully going through all the comments and for extensively rewriting the paper in response to the comments raised. I am for most parts happy with all the changes made.*

*One aspect however that is largely omitted is that of adsorption activation theory. There is one mention of it (without a reference) in the end of section 4 of the revised paper, which is clearly not enough. I am not requesting the data is reprocessed with the new theory (although that would be great to see), but not discussing it at all is an omission. For conditions of very low hygroscopicity, surface adsorption can be very important, even dominate CCN activity as showed by the studies cited in the previous round of reviews. There is even work on describing CCN activity of BC, using adsorption activation theory (e.g., https://doi.org/10.1029/2007JD008549).*

*I therefore would like to see a discussion on adsorption impacts in the introduction and the conclusions - stating explicitly that k-Kohler is used for simplicity/parameterization purposes, even if adsorption activation may be the underlying mechanism.*

Answer. A discussion on the absorption activation theory has been added to both the introduction and the conclusions, and the pertinent references added as kindly suggested by the Reviewers.

[revised manuscript text omitted]

**Comment [AF1]:** Even if not explicitly requested by the Reviewers, we added some more recent references.

Such a large variability of size distribution, morphology and chemical composition strongly impacts the reactivity of soot particles in the atmosphere and their propensity to evolve into CCN. Several studies exist that characterize the CCN activity of soot particles generated in the exhausts of laboratory flames (Lambe et al., 2015) and commercial burners like the miniCAST (Henning et al., 2012; Friebel et al., 2019). Soot particle aging
20 experiments are often performed in laboratory conditions that simulate the atmosphere and make use of flow reactors (Kotzick et al., 1997; Lambe et al., 2015; Zuberi et al., 2005) or atmospheric simulation chambers (Tritscher et al., 2011; Wittbom et al., 2014; Grimonprez et al., 2018). The hygroscopic properties of soot are generally determined at supersaturation conditions provided by instruments such as variable supersaturation condensation nuclei counters (VSCNC) or cloud condensation nuclei counters (CCNc). Overall, freshly emitted
25 soot particles are generally considered as poor CCN. However, several studies demonstrate that photochemical aging (Tritscher et al., 2011) or chemical aging that includes exposition to OH radicals (Zuberi et al., 2005; Lambe et al., 2015), to $O_3$ (Kotzick et al., 1997; Wittbom et al., 2014; Grimonprez et al., 2018) or to $NO_3$ radicals (Zuberi et al., 2005) under atmospheric relevant conditions can efficiently turn soot particles into CCN.

Köhler theory (Köhler, 1936) is widely used to describe the formation process of liquid cloud droplets at
30 supersaturation conditions. Köhler theory is entirely founded on equilibrium thermodynamics, and describes the change of the saturation vapor pressure of water induced by the curved surface of the nascent droplet and by the presence of solutes in the liquid phase. A number of recent implementations of Köhler theory have been used to describe the cloud droplet activation of wettable insoluble or partially soluble compounds. Among them, the adsorption activation theory describes the mechanism of droplet growth through multilayer
35 adsorption of water. The number of layers of adsorbed water molecules is calculated using Brunauer, Emmet and Teller isotherms (Henson, 2007), or alternatively the Frenkel-Halsey-Hill isotherms (Sorjamaa and Laaksonen, 2007). In this work, a simpler approach is chosen (Petters and Kreidenweis, 2007) that relies on a single parameter ($\kappa$) representation of the CCN activity to take into account the reduction of the water activity due to the presence of partially soluble components ($\kappa$-Köhler theory).
40     According to $\kappa$-Köhler theory, at thermodynamic equilibrium the supersaturation over an aqueous solution droplet $SS = SS(D, d_\text{p}, \kappa)$ as a function of the droplet diameter $D$, of the size of the seeding
45 particle $d_\text{p}$ and of the hygroscopic parameter $\kappa$ is given by:

**Comment [AF2]:** Reviewer#1, (16) and Reviewer#2, (1).

$$SS(D, d_{\mathrm{p}}, \kappa) = \frac{D^3 - d_{\mathrm{p}}^3}{D^3 - d_{\mathrm{p}}^3(1-\kappa)} \exp\left(\frac{A}{D}\right) - 1, \qquad A = \frac{4M_{\mathrm{w}}\sigma_{\mathrm{s/a}}}{R\,T\,\rho_{\mathrm{w}}} \qquad \text{Eq. (2)}$$

[revised manuscript text omitted]
_\mathrm{m} = \frac{C_\mathrm{c}(d_\mathrm{m})}{C_\mathrm{c}(d_\mathrm{pp})}\, d_\mathrm{pp} N_\mathrm{pp}^{\frac{\Gamma}{D_\mathrm{f}}} \qquad \text{Eq. (10)}$$

The dependence on $N_\mathrm{pp}$ can be removed by using the definition of $d_\mathrm{ve}$ in Eq. (5). Finally, Eq. (10) can be solved for $d_\mathrm{ve}$ to yield:

$$d_\mathrm{ve}(d_\mathrm{pp}, D_\mathrm{f}, d_\mathrm{m}) = d_\mathrm{pp} \left[ \frac{d_\mathrm{m}}{d_\mathrm{pp}} \frac{C_\mathrm{c}(d_\mathrm{pp})}{C_\mathrm{c}(d_\mathrm{m})} \right]^{\frac{D_\mathrm{f}}{3\Gamma}} \qquad \text{Eq. (11)}$$

$d_\mathrm{ve} = d_\mathrm{ve}(d_\mathrm{pp}, D_\mathrm{f}, d_\mathrm{m})$ can be calculated once size distribution and morphology of the aerosol are known using Eq. (11). $d_\mathrm{pp}$ and $D_\mathrm{f}$ are obtained from TEM imaging as explained below, while $d_\mathrm{m}$ is easy to access from SMPS measurements.

The functional analysis of Eq. (11) shown in Fig. 1 highlights the effect of the variability of (a) $d_\mathrm{pp}$ and (b) $D_\mathrm{f}$ on $d_\mathrm{ve}$ in a range important for soot particles. One of the DMA transfer functions obtained in this work (black solid line, see further below for details) is used as an example. In the cases investigated in this work, $d_\mathrm{ve} = d_\mathrm{ve}(d_\mathrm{pp}, D_\mathrm{f}, d_\mathrm{m})$ (colored dashed and dotted lines) is always shifted to smaller values and narrower than the original DMA transfer function. Fig. 1 shows the functional dependency of $d_\mathrm{ve}$ on (a) $d_\mathrm{pp}$ and (b) $D_\mathrm{f}$. As an example, the experimental $d_\mathrm{m}$ distribution of soot particles after size selection at 150 nm (black solid line, see section 3 for details on the experimental conditions) is compared to calculated $d_\mathrm{ve}$ (colored dashed and dotted lines). $d_\mathrm{m}$ is measured immediately before the droplet nucleation experiments in tandem SMPS configuration. In the cases investigated in this work, $d_\mathrm{ve} = d_\mathrm{ve}(d_\mathrm{pp}, D_\mathrm{f}, d_\mathrm{m})$ is always significantly shifted to smaller values and narrower than the original $d_\mathrm{m}$. Increasing $d_\mathrm{pp}$ from 10 nm up to 30 nm ($D_\mathrm{f}$ = 1.7) results in the main mode of $d_\mathrm{ve}$ increasing from 82.0 nm up to 117.0 nm. Similarly, increasing $D_\mathrm{f}$ from 1.6 up to 1.8 ($d_\mathrm{pp}$ = 20 nm) results in $d_\mathrm{ve}$ increasing from 93.5 nm up to 113.3 nm.

**Comment [AF7]:** Reviewer#1, (7).

[Figure]

**Fig. 1.** $d_m$ distribution of soot particles sampled from the kerosene jet diffusion flame at 130 mm HAB and size selected at 150 nm (black solid line) measured immediately before droplet nucleation experiments. Simulations of $d_{ve} = d_{ve}(d_{pp}, D_f, d_m)$ of soot particles having complex morphology according to Eq. (11).  For each series of simulations (colored dashed and dotted lines), (a) $d_m$ and $D_f$, or alternatively (b) $d_m$ and $d_{pp}$ are set as constant and the remaining parameter is varied in the range indicated in the legend.

> **Comment [AF8]:** Reviewer#1, (7). The legend and the caption of Fig. 1 have been corrected to remove the ambiguity between "DMA transfer function" and "$d_m$ after size selection".

**2.3. Taking into account the multimodality of $d_m$**

Because of the existence of multiple charges, one lognormal fit is generally not sufficient to describe the $d_m$ distributions. Therefore, with the aim of maintaining an analytical approach, an empirical multi-mode lognormal fit is used as a simple solution to describe the $d_m$ distribution of the aerosol particles injected in the CCNc:

$$\frac{dN(d_p)}{d d_p} = \sum_i \frac{N_i}{d_p \ln \sigma_{p,geo,i} \sqrt{2\pi}} e^{-\frac{[\ln d_p - \ln \mu_{p,geo,i}]^2}{2 \ln^2 \sigma_{p,geo,i}}}$$

Eq. (12)

where $\mu_{p,geo,i}$ and $\sigma_{p,geo,i}$ are the geometric mean and standard deviation of each mode.

> **Comment [AF9]:** Reviewer#1, (2), (3), (7) and 15(b-c). Eq. 12 is now moved to the Theory section in the dedicated Section 2.3., as its applicability is not only limited to ammonium sulfate.

[revised manuscript text omitted]

15   .

> **Comment [AF16]:** Reviewer#1, (2) and (3). Eq. 12, previously described here, is now moved to the end of the Theory section as its applicability is not only limited to ammonium sulfate. The text is updated accordingly.

For isolated, spherical and homogeneous particles, $d_{ve} = d_m = d_p$ (Eggersdorfer and Pratsinis, 2014; Sorensen, 2011). Although Fig. 3 shows a particularly favorable case, in this work it is found that $d_m = d_p$ is always true within 20% uncertainty. A summary of the parameters of the distributions is given in Table 1.

[Figure]

**Fig. 3.** (a) TEM image of size selected ammonium sulfate particles (black quasi-spherical particles) deposited on a Lacey mesh, 6500 magnification. (b) diameter of the particle projection $d_\mathrm{p}$ obtained from TEM measurements (red bars) and  single-mode lognormal fit (black dashed line). (c) electrical mobility diameter $d_\mathrm{m}$ obtained from SMPS measurements (black dots), and two-mode lognormal fit (red solid line) showing the two contributions (blue dotted and green dashed lines) according to Eq. (12).

The experimental activation data (black data points) and the calculated activation curves are shown in Fig. 4: $\mu_\mathrm{p,geo}$ and $\sigma_\mathrm{p,geo}$ are used as input parameters and obtained independently from SMPS ($d_\mathrm{m}$, red solid line) and TEM ($d_\mathrm{p}$, black dashed lines) measurements as shown in Fig. 3. Globally, the experimental data are in good agreement with the calculated curves.  In the fit, $\mu_\mathrm{\kappa,geo}$ is set as free parameter while $\sigma_\mathrm{\kappa,geo}$ is forced to unit value as no variability in the chemical composition of ammonium sulfate is expected. A detailed discussion on the impact of $\sigma_\mathrm{\kappa,geo}$ on the calculation of $F_\mathrm{a} = F_\mathrm{a}(SS)$ can be found in Annex A. The calculations from the independently obtained $d_\mathrm{m}$ and $d_\mathrm{p}$ result in  close $p(\kappa)$ having geometric mean $\kappa_\mathrm{SMPS}$ =  0.58±0.02 and $\kappa_\mathrm{TEM}$ = 0.61±0.02, both in excellent agreement with $\kappa$ = 0.61 found in the literature (Petters and Kreidenweis, 2007).

[Figure]

**Fig. 4.** Dry ammonium sulfate particles:  activation data obtained from CCNc experiments (black dots) and calculated $F_\mathrm{a} = F_\mathrm{a}(SS)$ using SMPS (red solid line) and TEM (black dashed lines) data as $d_\mathrm{ve}$.

[revised manuscript text omitted]

> **Comment [AF22]:** Reviewer#1, (2). Figure 5(d) has been updated to show the two-mode fit introduced with Eq. (12). The legend has been updated accordingly.

[Figure]

**Fig. 6. Soot sampled from the turbulent jet flame supplied with liquid kerosene at 130 mm HAB. (a) TEM picture of a soot aggregate showing $L_{2D}$ and $A_{2D}$. (b) $d_{pp}$ size distribution (red bars) and lognormal fit (black dashed line), $d_{pp}$ = 16.7 nm (mass equivalent $d_{pp}$ = 17.7 nm). (c) $\ln(N_{pp})$ vs. $\ln(L_{2D}/d_{pp})$ plot from which $D_f$ is obtained (100 projections). (d) normalized SMPS data**  **$d_m$ fit**  **size selection at 150 nm (black data points and**  **dashed line),**  **morphology-corrected $d_{ve}$ calculated using Eq. (11)**  **Eq. (12)** **.**

**Comment [AF23]:** Reviewer#1, (2). Figure 6(d) has been updated to show the two-mode fit introduced with Eq. (12). The legend has been updated accordingly.

**4.3. CCN activity of soot particles**

[revised manuscript text omitted]

**Comment [AF27]:** Reviewer#1, (16) and Reviewer#2, (1).

**Comment [AF28]:** Reviewer#1, (16) and Reviewer#2, (1).

**Comment [AF29]:** Reviewer#1, (1). Annex A has been added in response to the discussion over the sensibility of the fitting procedure.

[revised manuscript text omitted]